# PRISM: Synergizing Vision Foundation Models via Self-organized Expert Specialization

**Ying Tang** [1]   **Dong Li** [1]   **Youjia Zhang** [1]   **Zikai Song** [1]   **Junqing Yu** [1]   **Wei Yang** [1,†]

https://github.com/robotyingtang/PRISM-VFM

## Abstract

Unifying the complementary strengths of diverse Vision Foundation Models (VFMs) into a single efficient model is highly desirable but challenged by the negative transfer inherent in monolithic distillation. To address these feature conflicts, we introduce **PRISM**, a novel dual-stream Mixture-of-Experts (MoE) framework that synergizes VFMs via modular specialization. We propose a two-stage paradigm: (1) expertise deconstruction, where a teacher-conditional router guides experts to specialize in distinct representational subspaces to mitigate interference, followed by (2) dynamic recomposition, where the router learns to assemble these experts into tailored computational pathways for downstream tasks. Experiments on PASCAL-Context and NYUD-v2 show that **PRISM** establishes a new state of the art, validating that sparse, emergent specialization is a scalable approach for integrating diverse visual knowledge.

## 1. Introduction

A practical goal in computer vision is to consolidate multiple Vision Foundation Models (VFMs) into a single deployable student. Modern VFMs provide complementary cues, such as high-level semantics from CLIP (Radford et al., 2021), spatial structures from SAM (Kirillov et al., 2023), and fine-grained texture or correspondence from DINOv2 (Oquab et al., 2023); however, deploying them as a model ensemble is costly in memory, latency, and engineering complexity. The desired student should absorb these complementary visual dimensions into one parameter space while avoiding

the negative transfer caused by naive dense distillation.

However, compressing such heterogeneous knowledge into a unitary network introduces a fundamental optimization paradox. When shared parameters are forced to satisfy contradictory supervision signals, the model suffers from severe gradient conflict. For instance, while DINO encourages feature variance to distinguish local textures, CLIP often suppresses such variance to achieve semantic invariance. In standard dense architectures, these opposing gradient vector fields lead to destructive interference, causing the shared weights to collapse into a compromised **average** that fails to excel in either dimension.

To mitigate this interference, pioneering approaches such as SAK (Lu et al., 2025) adopt a divide-and-conquer strategy, assigning independent parameter branches to different teachers. While effective, this paradigm relies on the rigid assumption of hard boundaries, that visual knowledge can be explicitly sliced into disjoint domains. We argue that this is an oversimplification. In reality, visual knowledge exhibits **soft boundaries** with intricate overlaps. For example, both CLIP and DINO encode representations of a cat, yet they focus on different frequency bands (semantic identity vs. local texture) of the same entity. Static partitioning ignores this nuance, creating parameter redundancy and hindering the positive transfer of shared concepts. Consequently, a robust student requires a dynamic architecture capable of automatically perceiving the nature of the knowledge: sharing parameters for consensus and branching out only when functional conflicts arise.

In this work, we propose **PRISM** (**P**rojecting **R**epresentations into **I**ndependent **S**pecialized **M**odules), a framework that shifts from "manual partitioning" to self-organized specialization. Rather than explicitly assigning layers or modules to particular teachers or tasks, PRISM adopts a dual-stream gated Mixture-of-Experts (MoE) architecture for conflict-aware partial sharing. A *Universal Anchor stream* preserves stable shared representations and captures consensus knowledge, while a *Conditioned MoE stream* provides plasticity by routing tokens to sparse experts conditioned on layer, token, and

[1]School of Computer Science and Technology, Huazhong University of Science and Technology, Wuhan, China. Correspondence to: Wei Yang <weiyangcs@hust.edu.cn>.

*Proceedings of the 43rd International Conference on Machine Learning*, Seoul, South Korea. PMLR 306, 2026. Copyright 2026 by the author(s).

teacher/task context. Driven by context-modulated routing and locality-aware decorrelation, PRISM dynamically shares compatible knowledge, separates conflicting signals, and recombines specialized experts when knowledge partially overlaps, enabling emergent expert specialization while maintaining parameter efficiency.

Our contributions are summarized as follows:

- We conceptually reframe multi-teacher distillation as a dynamic consensus-conflict trade-off, identifying the limitations of static "hard boundary" partitioning in prior arts.

- We propose **PRISM**, a dual-stream MoE architecture that leverages Context-modulated routing to achieve implicit and emergent knowledge decomposition.

- To enable effective expert specialization, we introduce a locality-aware decorrelation mechanism that prevents semantic short-circuiting in shallow layers, acting as a critical inductive bias.

## 2. Related Work

**Knowledge Distillation of VFMs** As large-scale generalists, Vision Foundation Models (VFMs) demonstrate superior performance across diverse domains(e.g., vision (Song et al., 2026; Ye et al., 2025; Song et al., 2025) and multimodal tasks (Li et al., 2024a; Wang et al., 2026; Zhang et al., 2026)) with minimal tuning. Notable examples include CLIP (Radford et al., 2021) for vision-language understanding, DINOv2 (Oquab et al., 2023) for fine-grained representation learning, and SAM (Kirillov et al., 2023) for promptable segmentation. Many of them share attention-based designs (Song et al., 2022; 2023; 2024). Despite their efficacy, the substantial computational demands of these models have led to the widespread adoption of knowledge distillation (Vandenhende et al., 2021; Ye & Xu, 2023; Zong et al., 2024) for VFM compression and efficiency (Zhang & Yang, 2021; Ishihara et al., 2021; Yu et al., 2024).

Multi-teacher distillation has also been studied before the VFM era, and typically by aggregating soft targets or intermediate features from multiple relatively homogeneous teachers (Fukuda et al., 2017; You et al., 2017; Liu et al., 2020). More recently, research has shifted towards distilling multiple VFMs into a single student to synergize their heterogeneous strengths(Zhou et al., 2025; Ranzinger et al., 2024a; Shi et al., 2024). SAM-CLIP (Wang et al., 2024a) merges CLIP's semantic knowledge into SAM via continual learning. RADIO (Ranzinger et al., 2024b) distills CLIP, DINOv2, and SAM simultaneously to enhance downstream performance, a direction further refined by RADIOv2.5 (Heinrich et al., 2025), introducing improved training recipes and scaling laws for more robust feature

aggregation. Similarly, DUNE (Sarıyıldız et al., 2025) bridges the gap between 2D and 3D perception by distilling a universal encoder from heterogeneous teachers, while Theia (Shang et al., 2024) incorporates Depth Anything (Yang et al., 2024a) for robot learning. Furthermore, UNIC (Sariyildiz et al., 2024) utilizes multi-teacher distillation to consolidate specialized experts into a universal classification backbone. Most existing multi-teacher distillation methods (Ranzinger et al., 2024b; Shang et al., 2024) employ straightforward distillation into a dense backbone. We argue this leads to inherent feature interference, particularly between conflicting domains such as semantics and geometry. Unlike these approaches, PRISM adaptively transfers knowledge by retaining unique representation biases through a dynamic MoE mechanism to maximize strengths across multiple tasks.

**Multi-Task Learning Architectures** Multi-Task Learning (MTL)(Caruana, 1997) aims to train a single model capable of handling multiple tasks simultaneously (Ruder, 2017; Chen et al., 2020; Ishihara et al., 2021; Li et al., 2025; 2026). Research generally diverges into two categories: multi-task optimization (Chen et al., 2018; Guo et al., 2018; Kendall et al., 2018; Liu et al., 2021b;a; Li et al., 2024b) and model architecture design (Brüggemann et al., 2021; Ye & Xu, 2022a;b). In terms of architectures, existing methods are typically categorized into encoder-focused (Liu et al., 2019; Wang et al., 2025) and decoder-focused (Ye & Xu, 2022a) approaches. Recently, knowledge distillation (Zeng et al., 2026) has been integrated into MTL to bridge the gap between multi-task students and single-task teachers (Ranzinger et al., 2024b; Heinrich et al., 2025; Lu et al., 2025). Notably, Xu et al. (Xu et al., 2023) proposed directly distilling a small multi-task student from a large multi-task teacher.

Traditional MTL architectures typically rely on *static* sharing designs, which can be broadly categorized into encoder-focused (Liu et al., 2019) and decoder-focused (Ye & Xu, 2022a) approaches. Although such hard or partially shared parameterizations are simple and effective, static sharing can lead to negative transfer when task objectives conflict.

To improve flexibility, recent works have explored *dynamic* architectures based on Mixture-of-Experts (MoE). For instance, Mod-Squad (Chen et al., 2023) modularizes experts with an information-theoretic objective to balance cooperation and specialization, while TaskExpert (Ye & Xu, 2023) dynamically assembles representations through a memory-based MoE mechanism in the decoding stage. While promising, these methods do not explicitly address the semantic conflicts that may arise in multi-teacher distillation. In this regard, PRISM introduces conflict-aware routing for partial sharing, and remains complementary to standard MoE regularizers such as load balancing and routing entropy (Shazeer

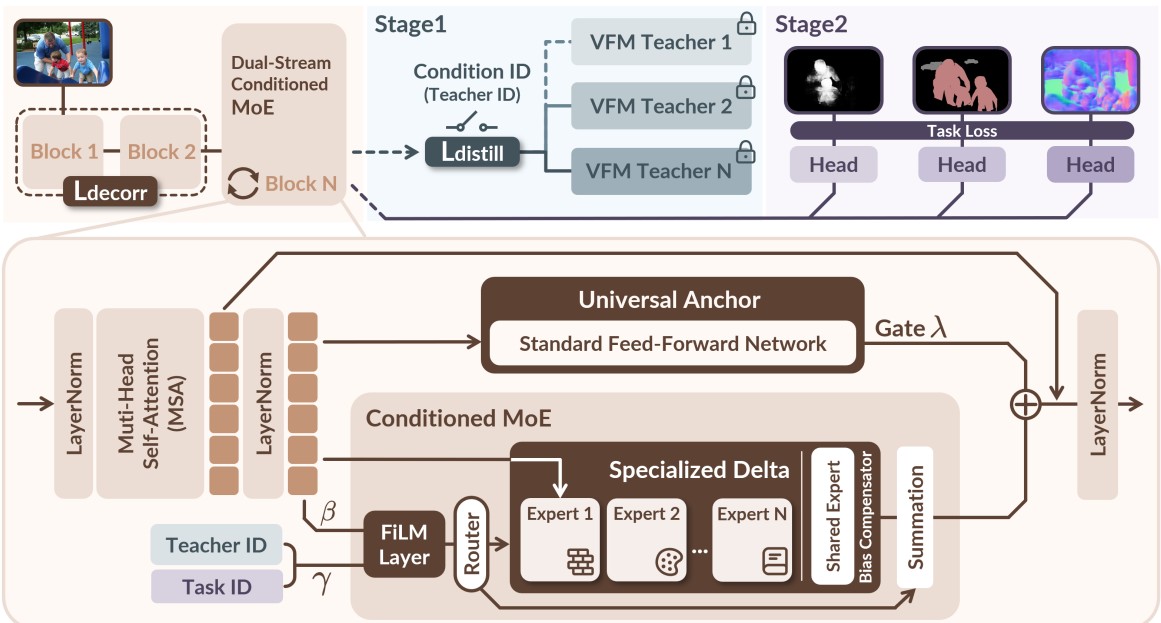

*Figure 1.* **Overview of the PRISM framework.** (Top) Two-Stage Training Pipeline. In Stage 1, the student (Dual-Stream Conditioned MoE) mimics multiple frozen VFM teachers. The Context ID (Teacher ID) conditions the routing, driving emergent knowledge decomposition. The $\mathcal{L}_{\text{decorr}}$ is applied to shallow layers to prevent rank collapse. In Stage 2, the model recombines experts for downstream tasks using the Task ID as context. (Bottom) Dual-Stream Architecture. The PRISM block replaces standard FFNs with two parallel paths: a Universal Anchor for shared consensus, and a Specialized Delta for conflict resolution. A FiLM-based Router modulates features based on context before dispatching tokens to sparse experts. A learnable gate $\lambda$ dynamically fuses the two streams.

et al., 2017; Fedus et al., 2022). It also differs from FiLM-based MoE variants such as MoFME (Zhang et al., 2024): in PRISM, FiLM conditions routing decisions rather than replacing expert computation.

In the specific context of distilling Vision Foundation Models (VFMs), recent approaches like RADIO (Ranzinger et al., 2024b) and SAK (Lu et al., 2025) have emerged. SAK represents the state-of-the-art in static architecture, employing a "Teacher-Agnostic Stem" plus "Teacher-Specific Adapters." While this reduces interference, its reliance on rigid, manual branching assumes hard boundaries between tasks, leading to structural redundancy when knowledge overlaps.

**Positioning PRISM.** Bridging these paradigms, PRISM proposes a dynamic encoder-focused architecture tailored for multi-teacher distillation. Unlike Mod-Squad which relies on loss constraints, or SAK which relies on static branches, PRISM leverages a context-modulated router to structurally resolve gradient conflicts. By treating experts as a basis for emergent knowledge decomposition, PRISM automatically shares parameters for consensus and branches out for conflicts token-by-token. This fine-grained mechanism solves the "Negative Transfer" problem more efficiently than rigid branching, offering a "best-of-both-worlds" solution between stability and plasticity.

## 3. Methodology

To resolve the gradient conflict in Multi-Teacher Distillation, we propose PRISM, a framework centered on the principle of "Decompose-then-Recombine". As illustrated in Figure 1, PRISM transforms a standard Vision Transformer (ViT) into a Dual-Stream Conditioned MoE architecture. PRISM introduces a dynamic trade-off mechanism: a Universal Anchor stream maintains stability for shared consensus, while a Specialized Delta stream provides plasticity for conflict resolution via context-modulated routing. The training proceeds in two stages: emergent knowledge decomposition from multiple VFM teachers (Stage 1), followed by knowledge recombination for downstream tasks (Stage 2), regularized by a locality-aware decorrelation loss.

### 3.1. Problem: Gradient Conflict in MTD

We formulate the task of Multi-Teacher Distillation (MTD) as a multi-objective optimization problem. Let $\mathcal{T} = \{T_1, T_2, \ldots, T_K\}$ denote a set of $K$ heterogeneous vision foundation models (VFMs), serving as teachers. The stu-

dent model $S$, parameterized by $\Theta$, aims to minimize the joint distillation loss over the training distribution $\mathcal{X}$:

$$\mathcal{L}_{\text{total}}(\mathbf{x}; \Theta) = \sum_{k=1}^{K} \gamma_k \mathcal{L}_{\text{distill}}(S(\mathbf{x}), T_k(\mathbf{x})) \qquad (1)$$

where $\gamma_k$ is a scalar coefficient balancing the contribution of each teacher.

**Gradient conflict in dense architectures.** Consider a standard dense layer (e.g., the FFN in a ViT block) with parameters $\theta \in \Theta$. During back-propagation, the parameter update $\Delta\theta$ is proportional to the aggregate gradient vector $\mathbf{g}_{\text{total}} = \sum_{k=1}^{K} \gamma_k \mathbf{g}_k$, where $\mathbf{g}_k = \nabla_\theta \mathcal{L}_{\text{distill}}(T_k)$. A fundamental optimization dilemma arises when teachers provide contradictory supervision for the same input features. Mathematically, this conflict manifests as a negative cosine similarity between task gradients:

$$\mathcal{C}_{i,j} = \cos(\mathbf{g}_i, \mathbf{g}_j) = \frac{\langle \mathbf{g}_i, \mathbf{g}_j \rangle}{\|\mathbf{g}_i\|\|\mathbf{g}_j\|} < 0 \qquad (2)$$

In a standard dense architecture where parameters are globally shared, severe conflict ($\mathcal{C}_{i,j} \ll 0$) leads to destructive interference, where $\mathbf{g}_i \approx -\mathbf{g}_j$. Consequently, the magnitude of the aggregate gradient $\|\mathbf{g}_{total}\|$ diminishes, and the optimization settles into a "compromised" equilibrium that is suboptimal for all tasks. We refer to this phenomenon as **gradient averaging**.

### 3.2. Motivation: Decomposition via Gradient Orthogonalization

To mitigate the gradient averaging dilemma, we argue that the student parameter space should distinguish between **consensus** and **conflict** components. Consensus knowledge, such as generic low-level structures, can be safely shared, whereas conflicting teacher-specific signals should update separated or weakly coupled parameters.

Specifically, for two teacher objectives $T_i$ and $T_j$ with conflicting gradients, i.e., $\cos(\mathbf{g}_i, \mathbf{g}_j) < 0$, PRISM encourages their routed effective gradients on each sparse expert $E_n$ to have small interaction:

$$\langle \tilde{\mathbf{g}}_{i,n}, \tilde{\mathbf{g}}_{j,n} \rangle \approx 0, \qquad (3)$$

where $\tilde{\mathbf{g}}_{i,n}$ denotes the gradient component that is actually routed to expert $E_n$ under teacher context $T_i$. This objective can be achieved either by reducing co-activation of the same expert or by making the residual gradients on co-active experts weakly aligned. It motivates our dual-stream design: the Universal Anchor preserves shared consensus, while the Conditioned MoE reduces effective interference through sparse, context-dependent dispatching.

### 3.3. Architecture: Dual-Stream Conditioned MoE

Guided by the orthogonality principle, we introduce the PRISM block to replace standard FFNs. Unlike SAK (Lu et al., 2025) which decomposes the network into a static "Teacher-Agnostic Stem" and "Teacher-Specific Adapters", PRISM relaxes this hard constraint into a continuous, dynamic trade-off between **Stability** and **Plasticity**. Formally, for an input $\mathbf{x}$, the block output is:

$$\mathbf{y} = \mathbf{x} + \underbrace{\lambda \cdot \mathcal{F}_{\text{anc}}(\text{LN}(\mathbf{x}))}_{\text{Stability Stream}} + \underbrace{(1 - \lambda) \cdot \mathcal{F}_{\text{moe}}(\mathbf{x}, c)}_{\text{Plasticity Stream}} \qquad (4)$$

where $\lambda \in [0, 1]$ is a learnable gating scalar. This gate allows the model to dynamically trade off between reusing common knowledge and invoking specialized experts. Empirical analysis (see Sec. 4.3) reveals that $\lambda$ naturally starts high in shallow layers (preferring stability) and decreases in deeper layers (favoring specialization), validating our hierarchical design hypothesis.

**Universal Anchor.** The first stream, $\mathcal{F}_{\text{anc}}$, is a standard dense MLP shared across all contexts. It captures task-agnostic, low-frequency patterns. We term this the **Stability Stream**, as it ensures the model maintains a robust optimization trajectory regardless of routing fluctuations.

**Specialized Delta.** The second stream, $\mathcal{F}_{\text{moe}}$, is a sparse Mixture-of-Experts. This stream offers a unified pool of experts accessed via Context-Modulated Routing. We term this the **Plasticity Stream**. By conditioning on context $c$, it provides the necessary plasticity to resolve gradient conflicts, allowing the model to emergently decide whether to invoke specific experts or combine them, handling Soft Boundaries that static methods miss.

**Network Instantiation.** We adopt ViT-Base (ViT-B/16) as the backbone. We replace the FFNs with Dual-Stream blocks in the 2nd, 5th, 8th, 11th layers. Each MoE layer contains $N = 15$ sparse experts with a Top-3 routing strategy, plus one internal shared expert for bias compensation.

### 3.4. Context-Modulated Routing Mechanism

To guide the Plasticity Stream, we employ a routing mechanism that directs token flow based on both image content and task intent.

**Context-Aware Feature Modulation.** Standard routers rely solely on input tokens $\mathbf{x}$, which is insufficient for distinguishing tasks with identical inputs. We inject the Context ID $c$ (Teacher ID in Stage 1, Task ID in Stage 2) via Feature-wise Linear Modulation (FiLM):

$$\hat{\mathbf{x}} = (1 + \gamma(c)) \odot \text{LayerNorm}(\mathbf{x}) + \beta(c) \qquad (5)$$

where $\gamma(c)$ and $\beta(c)$ are learned affine transformations. This

re-orients the feature space, enabling the router to make distinct decisions for CLIP vs. DINO contexts.

**Sparse Dispatching.** The router $G(\cdot)$ maps $\hat{\mathbf{x}}$ to expert weights via Top-K Softmax. The MoE output is:

$$\mathcal{F}_{\text{moe}}(\mathbf{x}, c) = E_{\text{shared}}(\mathbf{x}) + \sum_{i \in \text{TopK}} G(\hat{\mathbf{x}})_i E_i(\mathbf{x}) \quad (6)$$

The internal shared expert $E_{\text{shared}}$ captures common biases specific to the conflict subspace, further stabilizing the sparse routing.

### 3.5. Training Objectives and Strategy

**Locality-Aware Decorrelation Loss.** A critical prerequisite for effective expert specialization is the diversity of input tokens. However, in multi-teacher distillation, we identify a detrimental phenomenon we call "semantic short-circuiting". Driven by strong high-level supervision (e.g., from CLIP), the student model tends to bypass low-level feature extraction, causing shallow layers to prematurely converge to global semantic representations.

Mathematically, this manifests as rank collapse, where feature variance across tokens diminishes rapidly. If input tokens are homogeneous, the router lacks discriminative signals to dispatch them to different experts, leading to routing collapse. To counteract this, we explicitly inject a locality inductive bias by applying a Locality-Aware Decorrelation Loss ($\mathcal{L}_{\text{decorr}}$) to the shallow blocks. We penalize high cosine similarity between spatially distant pixels while preserving local correlations:

$$\mathcal{L}_{\text{decorr}} = \frac{1}{|\mathcal{P}|} \sum_{(i,j) \in \mathcal{P}} \max(0, \cos(\mathbf{z}_i, \mathbf{z}_j) - \epsilon) \cdot \mathbb{I}(d_{ij} > r) \quad (7)$$

where $\mathbf{z}_i$ is the feature vector at position $i$, $d_{ij}$ is the spatial Euclidean distance, and $r$ is a locality radius. This constraint forces shallow layers to encode rich, localized structural variations, providing high-quality, high-rank "raw materials" for the deep-layer experts.

**Training Strategy.** Our training pipeline consists of two distinct stages, as shown in Figure 1.

Stage 1: Emergent Knowledge Decomposition. The student learns from $K$ frozen VFM teachers. For each iteration, we randomly sample a teacher $T_k$ and use its ID as context $c$. The total loss is:

$$\mathcal{L}_{\text{stage1}} = \mathcal{L}_{\text{aux}} + \alpha \mathcal{L}_{\text{distill}} + \beta \mathcal{L}_{\text{decorr}} \quad (8)$$

Stage 2: Knowledge Recombination. In this stage, we fine-tune the model to adapt to downstream tasks while preserving the knowledge acquired in Stage 1. The total objective function is formulated as:

$$\mathcal{L}_{\text{stage2}} = \mu \mathcal{L}_{\text{distill}} + \sum_{t \in \mathbb{T}} w_t \mathcal{L}_t \quad (9)$$

where $\mathcal{L}_t$ is the task-specific loss for task $t$. The hyperparameter $\mu$ balances the distillation loss and the task losses, with a default value of 1.0, while $w_t$ adjusts the importance of each task. We set fixed $w_t$ values following the standard practice in MTL(Maninis et al., 2019; Kanakis et al., 2020).

## 4. Experiments

### 4.1. Experimental Setup

**Datasets and Protocol.** Following SAK (Lu et al., 2025), we perform Stage 1 pre-training on ImageNet-1k (Deng et al., 2009), followed by Stage 2 fine-tuning and evaluation on two standard multi-task benchmarks. **PASCAL-Context** (Mottaghi et al., 2014) covers five scene understanding tasks: Semantic Segmentation (SemSeg), Human Parsing (Parsing), Saliency, Surface Normal (Normal), and Boundary Detection (Boundary). **NYUD-v2** (Silberman et al., 2012) focuses on indoor scenes with four tasks: SemSeg, Depth Estimation, Normal, and Boundary.

**Evaluation Metrics.** We adopt standard metrics for each task: mean Intersection over Union (mIoU) for SemSeg and Parsing; maximal F-measure (maxF) for Saliency; optimal dataset scale F-measure (odsF) for Boundary; mean Error (mErr) for Normal Estimation; and RMSE for Depth Estimation. Lower values are better for mErr and RMSE. **VFM Teachers.** We distill knowledge from three frozen Vision Foundation Models (VFMs) with ViT-L backbones, unless otherwise specified: (1) DINOv2-L (Oquab et al., 2023): Provides robust local features and fine-grained correspondence. (2) CLIP-L (Radford et al., 2021): Offers high-level semantic understanding and language-aligned representations. (3) SAM-L (Kirillov et al., 2023): Contributes precise geometric and boundary cues from promptable segmentation. We extract teacher features from layers 5, 11, 17, and 23.

**Baselines.** We compare PRISM against three categories of methods: (1) **Task-Aware MTL Methods**, including encoder-decoder architectures like InvPT++ (Ye & Xu, 2024), TaskPrompter (Ye & Xu, 2022b), BFCI (Zhang et al., 2025), MLoRE (Yang et al., 2024b), and SEM (Huang et al., 2024); (2) **Unified Foundation Models**, such as RADIO/RADIOv2.5 (Ranzinger et al., 2024b; Heinrich et al., 2025), UNIC (Sariyildiz et al., 2024), and Theia (Shang et al., 2024), which distill multiple VFMs into a single backbone; and (3) **State-of-the-Art**, specifically **SAK** (Lu et al., 2025), which uses explicit architectural branching for multi-teacher distillation.

**Implementation Details.** We use ViT-B/16 as the backbone.

*Table 1.* **Comparison with state-of-the-art methods on PASCAL-Context (ViT-B backbone).** PRISM achieves the best overall performance ($\Delta_m = 2.29\%$) and outperforms SAK on all 5 tasks.

| Model | Semseg mIoU ↑ | Parsing mIoU ↑ | Saliency maxF ↑ | Normal mErr ↓ | Boundary odsF ↑ | $\Delta_m$ % ↑ |
|---|---|---|---|---|---|---|
| Single-task baseline | 80.25 | 70.54 | 84.54 | 13.57 | 74.22 | 0.00 |
| Multi-task baseline | 76.76 | 65.26 | 84.39 | 13.98 | 70.37 | -4.04 |
| InvPT (Ye & Xu, 2022a) | 77.33 | 66.62 | 85.14 | 13.78 | 73.20 | -2.28 |
| InvPT++ (Ye & Xu, 2024) | 76.95 | 66.89 | 85.12 | 13.54 | 73.30 | -1.92 |
| TaskPrompter (Ye & Xu, 2022b) | 79.00 | 67.00 | 85.05 | **13.47** | 73.50 | -1.24 |
| TaskExpert (Ye & Xu, 2023) | 78.45 | 67.38 | 84.96 | 13.55 | 72.30 | -1.73 |
| BFCI (Zhang et al., 2025) | 77.98 | 68.19 | 85.06 | 13.48 | 72.98 | -1.31 |
| MLoRE (Yang et al., 2024b) | 79.26 | 67.82 | **85.31** | 13.65 | 74.69 | -0.83 |
| RADIO (Ranzinger et al., 2024b) | 78.06 | 68.13 | 85.18 | 13.59 | 72.64 | -1.53 |
| RADIOv2.5 (Heinrich et al., 2025) | 81.75 | 71.49 | 81.26 | 16.10 | – | – |
| UNIC (Sariyildiz et al., 2024) | 75.90 | 62.85 | 81.84 | 15.78 | – | – |
| Theia (Shang et al., 2024) | 76.51 | 67.53 | 84.38 | 14.56 | 70.34 | -4.33 |
| SAK (Lu et al., 2025) | 81.88 | 74.30 | 84.79 | 14.02 | 74.09 | 0.83 |
| **PRISM (Ours)** | **82.20** | **75.34** | 84.81 | **13.47** | **75.92** | **2.29** |

*Table 2.* **Comparison with state-of-the-art methods on NYUD-v2 (ViT-B backbone).** PRISM surpasses SAK in semantic and depth estimation tasks.

| Model | Semseg mIoU ↑ | Depth RMSE ↓ | Normal mErr ↓ | Boundary odsF ↑ | $\Delta_m$ % ↑ |
|---|---|---|---|---|---|
| Single-task baseline | 51.15 | 0.5792 | 19.77 | 77.35 | 0.00 |
| Multi-task baseline | 49.27 | 0.5823 | 19.92 | 75.88 | -1.72 |
| InvPT (Ye & Xu, 2022a) | 50.30 | 0.5367 | 19.00 | 77.60 | 2.47 |
| InvPT++ (Ye & Xu, 2024) | 49.79 | 0.5318 | 18.90 | 77.10 | 2.40 |
| TaskPrompter (Ye & Xu, 2022b) | 50.40 | 0.5402 | 18.91 | 77.60 | 2.49 |
| ECS (Shoouri et al., 2023) | 50.46 | 0.5332 | 18.42 | 77.89 | 3.53 |
| BFCI (Zhang et al., 2025) | 51.14 | 0.5186 | 18.92 | 77.98 | 3.89 |
| TSP (Wang et al., 2024b) | 51.22 | 0.5301 | 18.78 | 76.90 | 3.26 |
| SEM (Huang et al., 2024) | 51.34 | 0.5222 | 18.95 | 77.60 | 3.67 |
| RADIO (Ranzinger et al., 2024b) | 55.03 | 0.5186 | 18.49 | 77.97 | 6.33 |
| RADIOv2.5 (Heinrich et al., 2025) | 57.19 | 0.4980 | 20.04 | – | – |
| UNIC (Sariyildiz et al., 2024) | 42.21 | 0.6172 | 22.78 | – | – |
| Theia (Shang et al., 2024) | 51.80 | 0.5367 | 19.70 | 76.08 | 1.83 |
| SAK (Lu et al., 2025) | 59.93 | 0.4942 | **17.60** | **78.60** | **11.11** |
| **PRISM (Ours)** | **60.22** | **0.4883** | 17.81 | 76.59 | 10.59 |

In stage 1, the model is trained for 30 epochs on ImageNet-1K. In stage 2, the model is trained for 40000 iterations on PASCAL-Context and NYUD-v2 using the AdamW optimizer. For Locality-Aware Decorrelation, we apply $\mathcal{L}_{decorr}$ to the first two layers, setting the hyperparameters in Eq. 8 as $\alpha = 0.9$ and $\beta = 0.1$.

### 4.2. Main Results

Table 1 and Table 2 summarize the quantitative comparisons on PASCAL-Context and NYUD-v2, respectively.

**Performance on PASCAL-Context.** PRISM establishes a new state-of-the-art, achieving an average improvement ($\Delta_m$) of 2.29%, clearly outperforming the previous best SAK (0.83%) and other strong competitors. Regarding superiority over SAK, PRISM surpasses it across all five tasks. Specifically, on Semantic Segmentation, PRISM achieves 82.20 mIoU (+0.32 vs. SAK), and on Human Parsing, it reaches 75.34 mIoU (+1.04). This suggests that our Context-Modulated Routing effectively preserves high-level semantic knowledge without rigid task-specific branching. Crucially, for geometric precision, PRISM improves Normal Estimation from 14.02 to 13.47 mErr and Boundary Detection from 74.09 to 75.92 odsF, indicating that emergent experts capture shared geometric structures more effectively than physically separated adapters.

**Performance on NYUD-v2.** On the indoor scene benchmark, PRISM remains highly competitive with SAK. It

*Table 3.* **Scaling Results on PASCAL-Context with ViT-L.** PRISM obtains the best overall $\Delta_m$.

| Model | Semseg mIoU ↑ | Parsing mIoU ↑ | Saliency maxF ↑ | Normal mErr ↓ | Boundary odsF ↑ | $\Delta_m$ % ↑ |
|---|---|---|---|---|---|---|
| Single-task baseline | 81.61 | 72.77 | 83.80 | 13.87 | 75.24 | 0.00 |
| SAK | 84.01 | 76.99 | 84.65 | 13.82 | **76.27** | 2.30 |
| **PRISM** | **84.34** | **77.83** | **84.67** | **13.43** | 76.23 | **3.16** |

improves Semantic Segmentation (60.22 vs. 59.93 mIoU) and Depth Estimation (0.4883 vs. 0.4942 RMSE), indicating effective transfer of semantic and geometry-aware knowledge. Meanwhile, SAK retains an advantage on Surface Normal and Boundary Detection, likely because its dedicated adapters provide stronger task-specific locality for indoor geometric and high-frequency cues. This reflects a dataset-dependent balance between flexible cross-teacher recombination and specialized local adaptation.

**Scaling to ViT-L.** We further evaluate whether PRISM remains effective with a larger student backbone. As shown in Table 3, following the same ImageNet-1K pretraining and PASCAL-Context fine-tuning protocol as SAK, PRISM with ViT-L achieves a $\Delta_m$ of **3.16%**, compared with **2.30%** for SAK. This suggests that the proposed decomposition-and-recombination mechanism is not limited to ViT-B, but continues to improve conflict-aware partial sharing at a larger model scale.

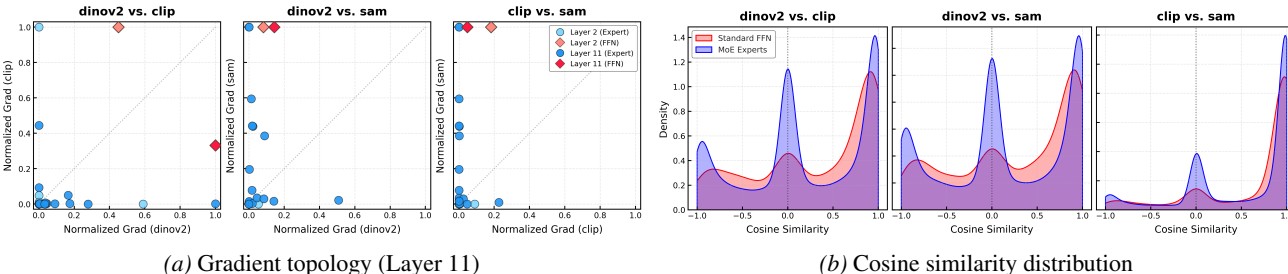

*(a)* Gradient topology (Layer 11)        *(b)* Cosine similarity distribution

*Figure 2.* Visualization of effective VFM conflict reduction. (a) The joint distribution of gradient norms. Magnitudes are independently normalized to $[0, 1]$ to compare geometric tendencies. Standard FFNs (diamonds) show broad simultaneous updates from multiple VFMs, indicating dense parameter entanglement. In contrast, MoE experts (circles) form an L-shaped topology, suggesting that many sparse experts are predominantly updated by one VFM condition. (b) The density of cosine similarities between VFM gradients. Sparse experts (blue) concentrate around zero, indicating reduced effective interaction, whereas the shared FFN (red) exhibits broader correlations caused by simultaneous multi-teacher updates.

### 4.3. Gradient-Based Conflict Analysis

To validate whether PRISM resolves the optimization conflict among different Vision Foundation Models (VFMs), we analyzed the gradient dynamics. Let $\theta$ denote the parameter vector (either in Sparse Experts or the Shared FFN). We define $g_v = \nabla_\theta \mathcal{L}_v$ as the gradient derived from the distillation loss $\mathcal{L}_v$ corresponding to a specific VFM $v \in \{\text{DINOv2}, \text{CLIP}, \text{SAM}\}$.

**Gradient Topology.** We visualized the joint distribution of gradient $L_2$-norms for VFM pairs (Figure 2a). To compare geometric tendencies despite scale disparities, we applied independent normalization, scaling gradients of each module to $[0, 1]$ by their respective maximums.

The shared FFN parameters (red diamonds) receive broad simultaneous updates from multiple VFMs. They are often distributed in the interior region rather than concentrated near a single axis, indicating that dense parameters are jointly affected by heterogeneous teacher signals. Such simultaneous updates create the optimization condition under which gradient averaging can occur. In contrast, the MoE experts (blue circles) exhibit an L-shaped topology. Many experts lie close to one axis, meaning that their dominant update comes from one VFM condition while the update from another is weak. For pairs involving SAM, some experts deviate from the axes due to partially shared structural cues, but they remain more separated than the dense FFN parameters. This pattern suggests that PRISM decomposes VFM knowledge into sparse expert subspaces and reduces effective cross-teacher interference.

**Cosine Similarity.** We further quantified interference by calculating the neuron-level cosine similarity: $\text{Cos}(\theta) = (g_{v_A} \cdot g_{v_B})/(\|g_{v_A}\|\|g_{v_B}\| + \epsilon)$.

Figure 2b further shows that sparse experts and shared FFNs follow different interaction patterns. The shared FFN (red)

*Table 4.* **Component Analysis on PASCAL-Context (ViT-S Student).** We evaluate the impact of the Dual-Stream design, FiLM routing, and Decorrelation Loss. The full PRISM model achieves the best trade-off.

| Configuration | Semseg
mIoU ↑ | Parsing
mIoU ↑ | Saliency
maxF ↑ | Normal
mErr ↓ | Boundary
odsF ↑ |
|---|---|---|---|---|---|
| (1) w/o MoE | 78.04 | 67.55 | 84.97 | 14.31 | 70.16 |
| (2) w/o Anchor | 78.46 | 67.98 | 84.41 | 14.65 | 69.08 |
| (3) w/o FiLM | 78.87 | **69.26** | 84.64 | 14.46 | 70.63 |
| (4) w/o $\mathcal{L}_{decorr}$ | **79.80** | 61.80 | **85.50** | 14.32 | **70.86** |
| **(5) PRISM (Full)** | 79.19 | 69.25 | 85.01 | **14.28** | 70.78 |

has a broad distribution, reflecting the fact that the same dense parameters are updated by multiple VFMs. In contrast, the sparse experts (blue) place more mass near zero cosine similarity, indicating that routed teacher-specific updates become weakly interacting on many expert parameters. Positive modes correspond to compatible teacher signals that can be safely shared, while negative modes usually occur with highly imbalanced gradient magnitudes, where one teacher provides the dominant update. Overall, the transition from broad dense coupling to sparse, near-axis expert updates supports the intended mechanism of PRISM: reducing effective interference while preserving useful shared signals.

### 4.4. Ablation Study

To dissect the contribution of each component in PRISM, we conduct comprehensive ablation studies using a ViT-S student and ViT-B teachers. Stage 1 is trained on 10% of ImageNet-1k, followed by fine-tuning on PASCAL-Context.

**Impact of Dual-Stream Architecture.** We first validate the necessity of the hybrid design. As shown in Row 1 in Table 4, the dense baseline (w/o MoE) struggles with capacity, yielding the lowest segmentation performance at 78.04 mIoU. Critically, removing the Universal Anchor (Row 2) results in the worst surface normal estimation error

*Table 5.* **Controlled Comparison on PASCAL-Context (ViT-S Student).** Wide ViT-S controls for static dense capacity scaling, while vanilla MoE controls for generic sparse routing.

| Model | Semseg mIoU ↑ | Parsing mIoU ↑ | Saliency maxF ↑ | Normal mErr ↓ | Boundary odsF ↑ |
|---|---|---|---|---|---|
| vanilla MoE | 72.96 | 61.77 | 83.34 | 15.03 | 66.73 |
| Wide ViT-S | 78.15 | 67.71 | 84.88 | **14.24** | 70.35 |
| **PRISM** | **79.19** | **69.25** | **85.01** | 14.28 | **70.78** |

*Table 6.* **Cost-performance Comparison on ViT-S.** PRISM-lite keeps the same PRISM design but reduces the expert MLP ratio from 4.0 to 1.0.

*(a)* Efficiency.

| Model | Total Params (M) | Active Params (M) | GFLOPs | Latency (ms) |
|---|---|---|---|---|
| SAK ViT-S | 26.40 | 26.40 | 53.20 | 20.25 |
| **PRISM-lite** | 48.05 | 38.06 | 57.51 | 47.51 |

*(b)* Performance on PASCAL-Context.

| Model | Semseg mIoU ↑ | Parsing mIoU ↑ | Saliency maxF ↑ | Normal mErr ↓ | Boundary odsF ↑ | $\Delta_m$ % ↑ |
|---|---|---|---|---|---|---|
| SAK ViT-S | 78.66 | 68.46 | 84.66 | **14.33** | **70.28** | 0.43 |
| **PRISM-lite** | **78.97** | **69.60** | **84.71** | 14.39 | 69.73 | **0.61** |

(14.65) among all configurations. This specific degradation confirms that while sparse experts handle task divergence, the shared anchor is indispensable for maintaining coherent structural and geometric representations.

**Effectiveness of Context-Modulated Routing.** Next, we investigate the routing mechanism. Replacing the FiLM-based router with simple task embedding concatenation (Row 3) leads to sub-optimal routing, dropping Semantic Segmentation by 0.32 mIoU and increasing normal error compared to the full model. This indicates that static task IDs are insufficient; the router requires the input-conditional feature recalibration provided by FiLM to align features with expert subspaces dynamically.

**Necessity of Locality-Aware Decorrelation.** The decorrelation loss is critical for balanced specialization. Removing $\mathcal{L}_{decorr}$ keeps several coarse metrics strong, but causes Human Parsing to drop sharply from 69.25 to 61.80 mIoU. Human Parsing is particularly sensitive because it requires fine-grained separation of adjacent body parts with thin and ambiguous boundaries. Additional routing analysis in Appendix C shows that $\mathcal{L}_{decorr}$ reorganizes Parsing away from geometry-dominated routing and toward more task-appropriate semantic/boundary sharing.

**Beyond Capacity Scaling and Plain MoE.** To isolate the source of PRISM's improvements, we compare it with two controlled baselines that account for potential confounding factors in capacity and sparsity. First, we construct an iso-FLOPs **Wide ViT-S** baseline by increasing the FFN expansion ratio to $5\times$, matching the active computation of PRISM's Universal Anchor, shared experts, and Top-3 routed experts. We initialize this widened model with **Net2Net tiling** (Chen et al., 2015) to ensure stable convergence, thereby testing whether the gains simply come from static dense capacity scaling. Second, we compare with a vanilla sparse MoE of comparable active capacity, obtained by removing PRISM-specific components, including the Universal Anchor, FiLM-conditioned routing, shared expert, and dual-stream fusion. This control examines whether the gains arise merely from introducing sparse expert routing. As shown in Table 5, PRISM consistently outperforms both baselines, suggesting that its advantage comes from conflict-aware dual-stream routing and dynamic knowledge disentanglement rather than increased dense capacity or

generic MoE sparsity alone.

**Cost-Performance Trade-off.** PRISM introduces extra offline cost because Stage 1 requires forwarding frozen VFM teachers. However, this cost is paid only during distillation, while inference uses a single student without running multiple VFMs. To examine whether the design remains useful under a lighter setting, we evaluate **PRISM-lite**, which keeps the same dual-stream architecture, FiLM routing, and losses, but reduces the expert MLP ratio from 4.0 to 1.0. As shown in Table 6, PRISM-lite achieves higher overall $\Delta_m$ than SAK under comparable GFLOPs, although sparse dispatch introduces higher latency.

### 4.5. Analysis of Emergent Specialization

To verify our "Decompose-then-Recombine" hypothesis and validate the existence of soft boundaries, we probe the routing dynamics of the MoE layer (Layer 11) on the validation set by visualizing the expert activation probability $P(\text{Expert}_i|\text{Condition})$ in Figure 3.

**Phase 1: Emergent Decomposition.** We first observe the emergence of teacher specialization and partial consensus in the routing patterns. The experts partition into distinct clusters: $E_{14}$ acts as a semantic specialist dominated by CLIP (20.7%), while $E_2$ serves as a geometric specialist dominated by SAM (14.2%), confirming that PRISM successfully separates conflicting supervision signals into more specialized parameter subspaces. Crucially, the router also identifies experts representing partial consensus, such as $E_{12}$, which is co-activated by DINOv2 and CLIP for shared texture semantics but ignored by SAM. This highlights a fundamental advantage over SAK. While SAK's static architecture forces a rigid dichotomy between global consensus and task specificity, often leading to redundant learning of shared features, PRISM naturally identifies these soft boundaries. By consolidating shared knowledge into specific experts like $E_{12}$, our method achieves greater parameter efficiency.

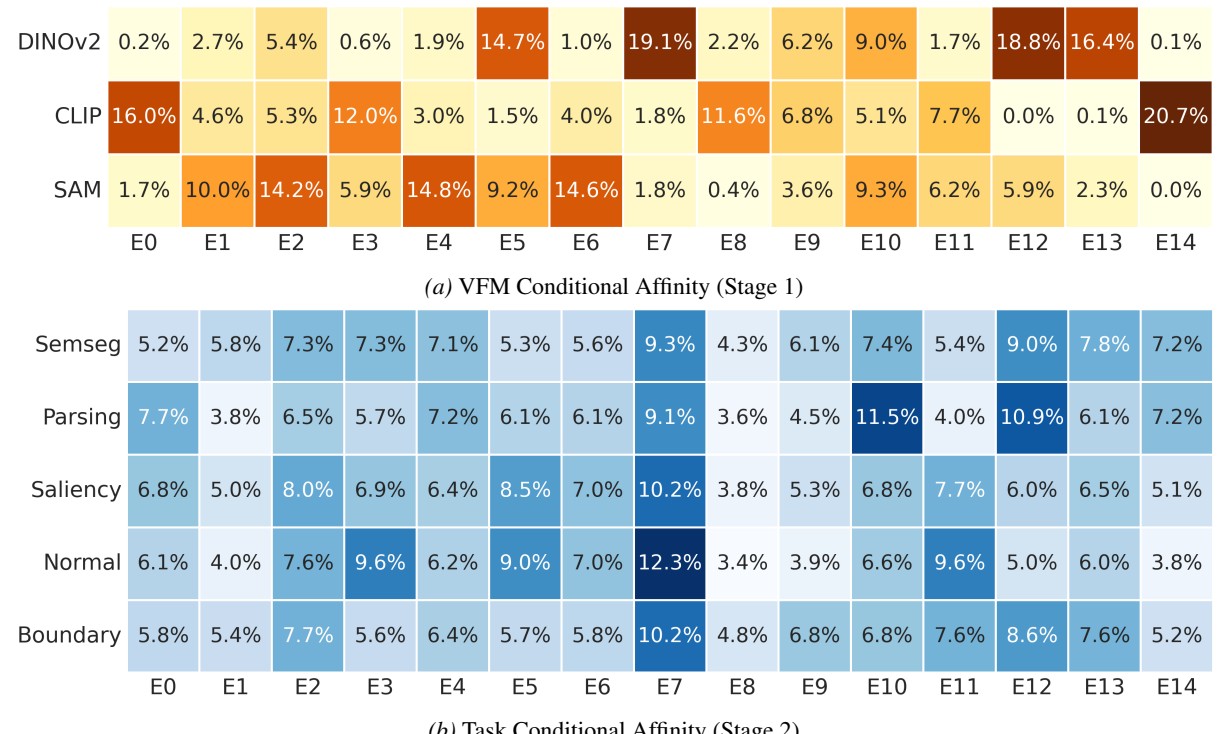

*(a)* VFM Conditional Affinity (Stage 1)

*(b)* Task Conditional Affinity (Stage 2)

*Figure 3.* **Visualization of Emergent Knowledge Decomposition and Recombination at Layer 11. (a)** When conditioned on Teachers, experts show clear specialization (e.g., $E_{14}$ for CLIP, $E_7$ for DINOv2), supporting effective conflict reduction through expert specialization. **(b)** When conditioned on Tasks, the router *recombines* these experts. For example, SemSeg (Semantic) utilizes both DINO-experts and CLIP-experts, whereas Normal (Geometric) ignores CLIP-experts ($E_{14}$) in favor of DINO-experts ($E_7$).

**Phase 2: Knowledge Recombination.** In the second stage, we analyze how these specialized primitives are utilized for downstream tasks. Unlike the sparse activation seen during teacher training, tasks exhibit composite routing patterns that recombine knowledge. Semantic Segmentation recruits a hybrid coalition, activating both the partial consensus expert $E_{12}$ (9.0%) for spatial correspondence and the semantic expert $E_{14}$ (7.2%) for categorical reasoning. Conversely, Surface Normal Estimation heavily relies on texture-aware experts while significantly reducing reliance on the CLIP-specific $E_{14}$ to 3.8%. This validates that PRISM moves beyond rigid partitioning to decompose knowledge into granular primitives, which are then dynamically recombined based on task needs.

**Learnable Stability-Plasticity Trade-off.** Finally, we track the trajectory of the learnable gate $\lambda$ across layers to understand the adaptive balance between shared and specialized features. In shallow layers such as Layer 2, $\lambda$ converges to approximately $0.7$, retaining high reliance on the Universal Anchor for stability. In deeper layers such as layers 5, 8 and 11, $\lambda$ shifts to approximately $0.5$, indicating a greater dependence on Specialized Experts. This trend aligns with the intuition that conflicting semantic concepts primarily arise in deeper representations, necessitating stronger expert

intervention to resolve interference.

## 5. Conclusion

In this work, we propose **PRISM**, a dynamic framework that addresses the optimization paradox in multi-teacher distillation through a **"Decompose-then-Recombine"** paradigm. By introducing a Dual-Stream Conditioned MoE, our architecture reduces effective cross-teacher interference through sparse expert specialization while preserving shared consensus via a Universal Anchor. Extensive experiments demonstrate that this emergent specialization leads to state-of-the-art performance across dense prediction benchmarks, effectively harmonizing the distinct strengths of heterogeneous Vision Foundation Models. Diagnostic analyses further show that context-modulated routing captures soft boundaries between knowledge domains and supports dynamic knowledge recombination. One remaining trade-off lies in balancing semantic abstraction and geometric locality: PRISM learns a shared pool of specialized experts for heterogeneous VFM knowledge, but the optimal allocation between high-level semantic cues and local geometric cues can vary across datasets and tasks. Future work may explore task-adaptive routing or lightweight local refinements to further tune this balance.

## Acknowledgements

This work was supported by the National Natural Science Foundation of China (Nos. 62272184 and 62402189), the China Postdoctoral Science Foundation (Nos. 2024M751012, 2025T180429, and GZC20230894), the Postdoctor Project of Hubei Province (No. 2024HBB-HCXB014), the Natural Science Foundation of Hubei Province (No. JCZRMS202600758), and the CIPS-SMP-Zhipu Large Model Fund (No. CIPS-SMP20250306). The computational work was performed on the high-performance computing platform at Huazhong University of Science and Technology.

## Impact Statement

This paper presents work whose goal is to advance the field of Machine Learning. There are many potential societal consequences of our work, none which we feel must be specifically highlighted here.

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

# A. Effective Cross-Teacher Interaction

In the main paper, we use gradient topology and cosine-similarity distributions to show that PRISM reduces effective cross-teacher interference. Here we provide a more detailed pair-by-layer-by-checkpoint analysis. This analysis directly measures whether two teacher conditions update the same sparse experts, how much their token-level routing overlaps, and whether the remaining co-active experts receive aligned or conflicting gradients.

For teacher contexts $a$ and $b$, we measure the effective sparse-path interaction at layer $l$ as

$$I_l^{(a,b)} = \mathbb{E}\left[ \mathbf{1}(E_l^a \cap E_l^b \neq \emptyset) \cdot \frac{1}{|E_l^a \cap E_l^b|} \sum_{e \in E_l^a \cap E_l^b} \cos(g_{l,e}^a, g_{l,e}^b) \right]. \tag{10}$$

This quantity is zero when no sparse expert is co-activated, and otherwise measures the residual gradient interaction on the actually shared sparse parameters.

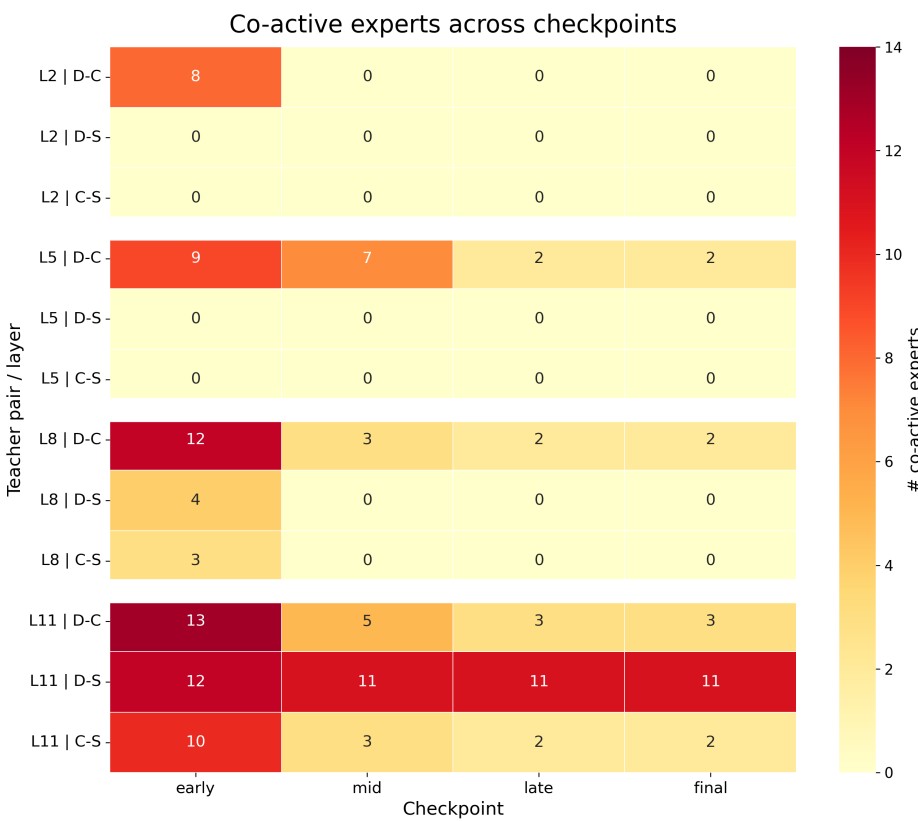

*Figure 4.* **Co-active sparse experts across checkpoints.** Each cell shows the number of experts that are activated under both teacher conditions for a teacher pair at a given MoE layer. Co-activation drops sharply during training: pair/layer cases with zero co-active experts increase from $4/12$ at the early checkpoint to $7/12$ by mid/late/final. Earlier layers become exactly separated more often, while deeper layers retain limited overlap, most notably DINOv2–SAM at Layer 11.

Figure 4 first shows the number of co-active experts across checkpoints. Co-activation drops sharply during training: the number of pair/layer cases with zero co-active sparse experts increases from $4/12$ at the early checkpoint to $7/12$ by the mid, late, and final checkpoints. This supports the intended sparse-path separation mechanism. Earlier layers become exactly separated more often, while deeper layers preserve limited overlap, most notably DINOv2–SAM at Layer 11, which is consistent with their partially shared structural cues.

Figure 5 provides a representative DINOv2–CLIP case. The top panel reports the mean cosine similarity on co-active sparse experts, while the bottom panel reports the cosine similarity on the shared branch. In Layer 5, the sparse-expert cosine decreases from $0.235$ at the early checkpoint to $0.189$ at the middle checkpoint, and then becomes negative at late/final checkpoints $(-0.167/-0.166)$. Cross markers indicate checkpoints where no co-active sparse experts exist, i.e., the effective sparse interaction is zero.

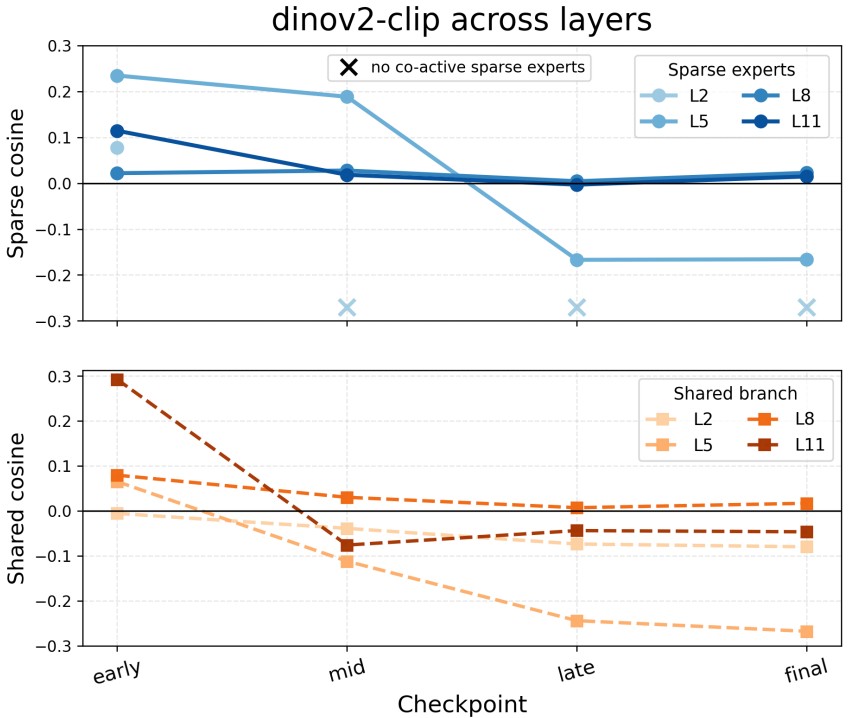

*Figure 5.* **Representative gradient-interaction trend for DINOv2–CLIP across checkpoints from the same training run.** Top: mean cosine similarity of gradients received by co-active sparse experts at different checkpoints. Bottom: cosine similarity on the shared branch. Cross markers denote checkpoints where no co-active sparse experts exist, i.e., effective sparse interaction is zero in practice. A representative example is Layer 5, where the sparse-expert cosine changes from $0.235$ at early to $0.189$ at mid, then becomes negative at late/final $(-0.167/-0.166)$, indicating substantially reduced cross-teacher interference.

Figure 6 gives the complete characterization over all three teacher pairs, four MoE layers, and four checkpoints. PRISM reduces sparse-path interference through two complementary regimes. In many cases, routing overlap and co-activation decrease directly. In cases where overlap persists, such as some deeper DINOv2–SAM settings, the residual sparse-expert gradient cosine remains close to zero. Meanwhile, the shared expert retains small but nonzero compatible gradients, indicating structured decomposition rather than indiscriminate suppression of all shared information.

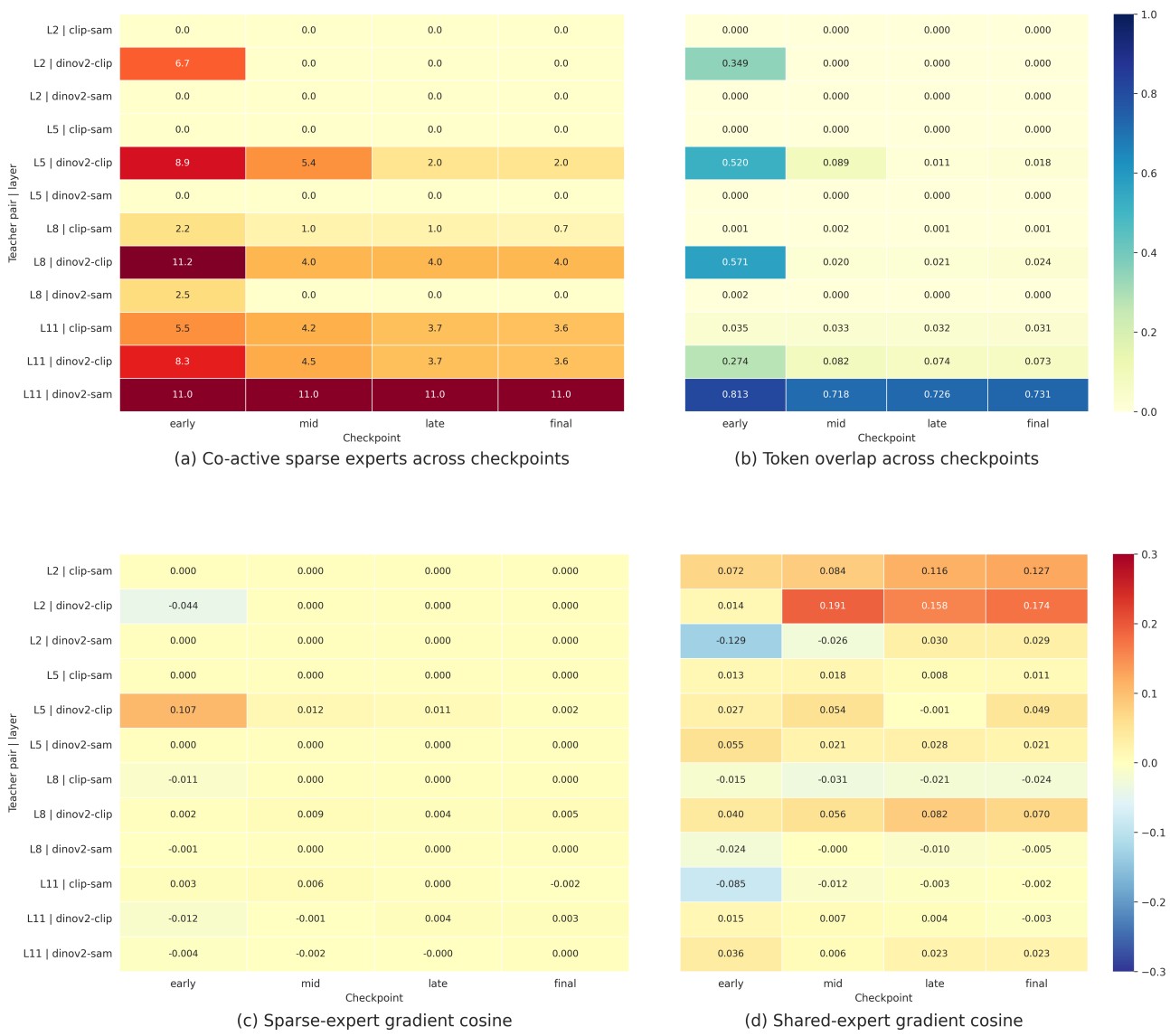

(a) Co-active sparse experts across checkpoints

(b) Token overlap across checkpoints

(c) Sparse-expert gradient cosine

(d) Shared-expert gradient cosine

*Figure 6.* **Pair × layer × checkpoint characterization of cross-teacher interaction.** We show (a) co-active sparse experts, (b) token overlap, (c) gradient cosine on the actually co-active sparse experts, and (d) gradient cosine on the shared expert. PRISM reduces sparse-path interference either by decreasing routing overlap/co-activation or, when overlap persists, by driving the residual sparse-expert gradient cosine near zero. The shared expert retains small but nonzero compatible gradients, indicating structured decomposition rather than global suppression.

## B. Stability Across Random Seeds

We next examine whether expert specialization is a stable phenomenon or a seed-specific artifact. Since MoE expert indices are permutation-invariant, we align non-reference seeds to the first seed with Hungarian matching before comparison. Figure 7 shows that the same qualitative teacher–expert affinity structure consistently re-emerges across random seeds.

In early and middle layers, teacher-preferred partitions are sharp: different teachers activate different expert groups

Teacher–Expert Affinity Across Seeds

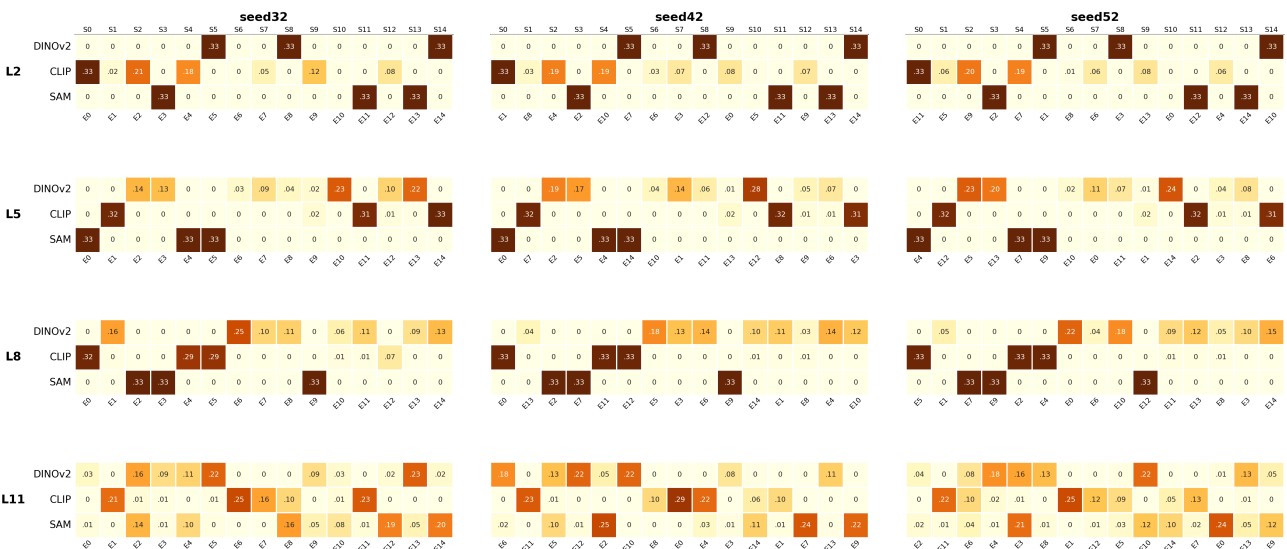

*Figure 7.* **Teacher–expert affinity across random seeds during the decompose stage.** Each heatmap shows the routing-frequency affinity matrix for one MoE layer. Non-reference seeds are aligned to the first seed using Hungarian matching. The top axis shows aligned comparison slots $S_k$, while the bottom axis shows the original expert IDs $E_j$ in each seed. Cells with zero or negligible affinity are left blank for readability. Earlier and middle layers exhibit sharper teacher-preferred partitioning, whereas the deepest layer is more mixed/shared; importantly, the same qualitative specialization structure consistently re-emerges across seeds up to permutation.

after alignment. In the deepest layer, routing becomes more mixed and shared, which is expected because high-level representations contain more overlapping semantic and structural information. This seed-level stability supports that the observed specialization is a robust outcome of the PRISM training objective and architecture.

## C. Effect of Locality-Aware Decorrelation on Human Parsing

The component ablation in the main paper shows that removing $\mathcal{L}_{decorr}$ causes a large drop in Human Parsing. This section provides additional evidence explaining why this task benefits strongly from locality-aware decorrelation. Human Parsing requires fine-grained separation among adjacent body parts, where local boundaries are thin and often ambiguous. Therefore, a routing pattern dominated by coarse geometry can hurt part-level discrimination. We conduct this analysis as an additional diagnostic evaluation to localize the effect of $\mathcal{L}_{decorr}$; while its absolute mIoU values may differ from the main ablation table due to the diagnostic protocol, all comparisons within this section use the same protocol.

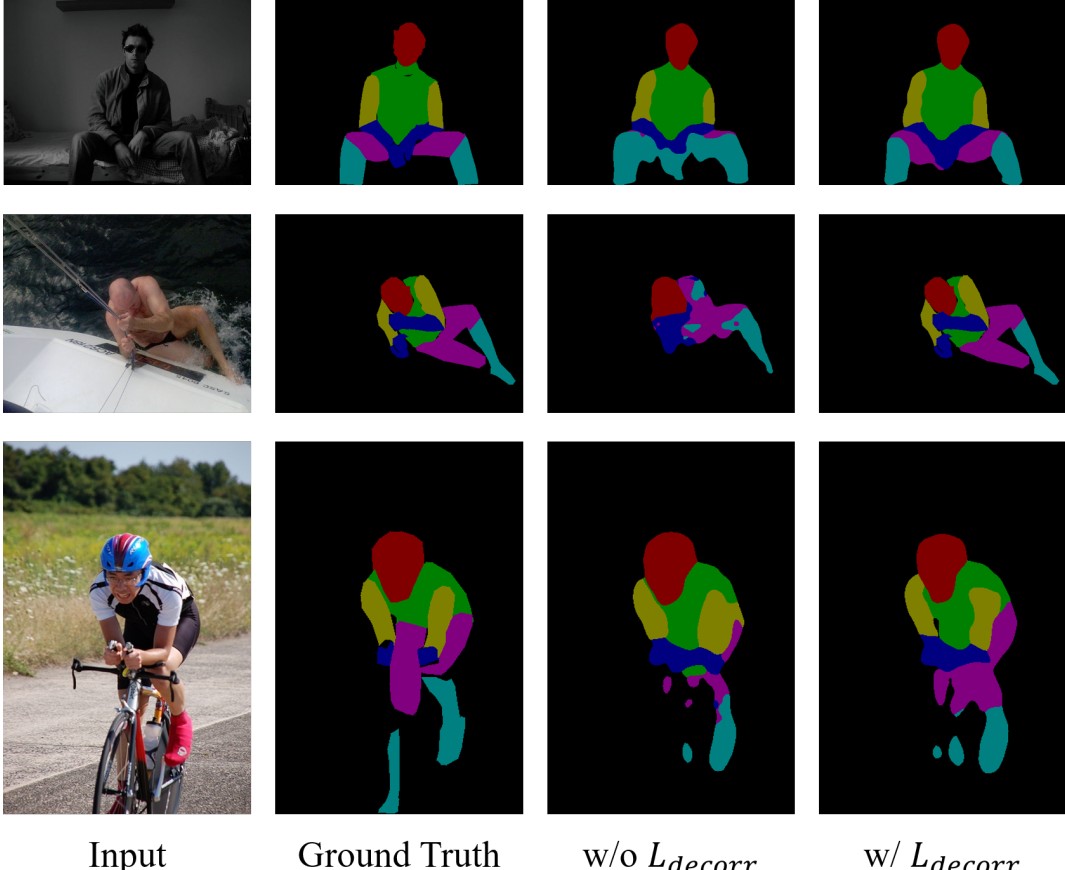

| Input | Ground Truth | w/o $L_{decorr}$ | w/ $L_{decorr}$ |

*Figure 8.* **Qualitative comparison on Human Parsing.** We show three representative examples for illustration. Compared with the variant without $\mathcal{L}_{decorr}$, adding $\mathcal{L}_{decorr}$ generally yields more coherent part layouts and reduces fragmented predictions, especially around lower-body regions and articulated limbs. This is consistent with the improvements in boundary-band evaluation and the largest per-class gains on limb categories.

Figure 8 shows representative qualitative examples. With $\mathcal{L}_{decorr}$, predictions become more coherent and less fragmented, especially around lower-body regions and articulated limbs. This qualitative trend matches the quantitative improvements in Tables 7, 8, and 9.

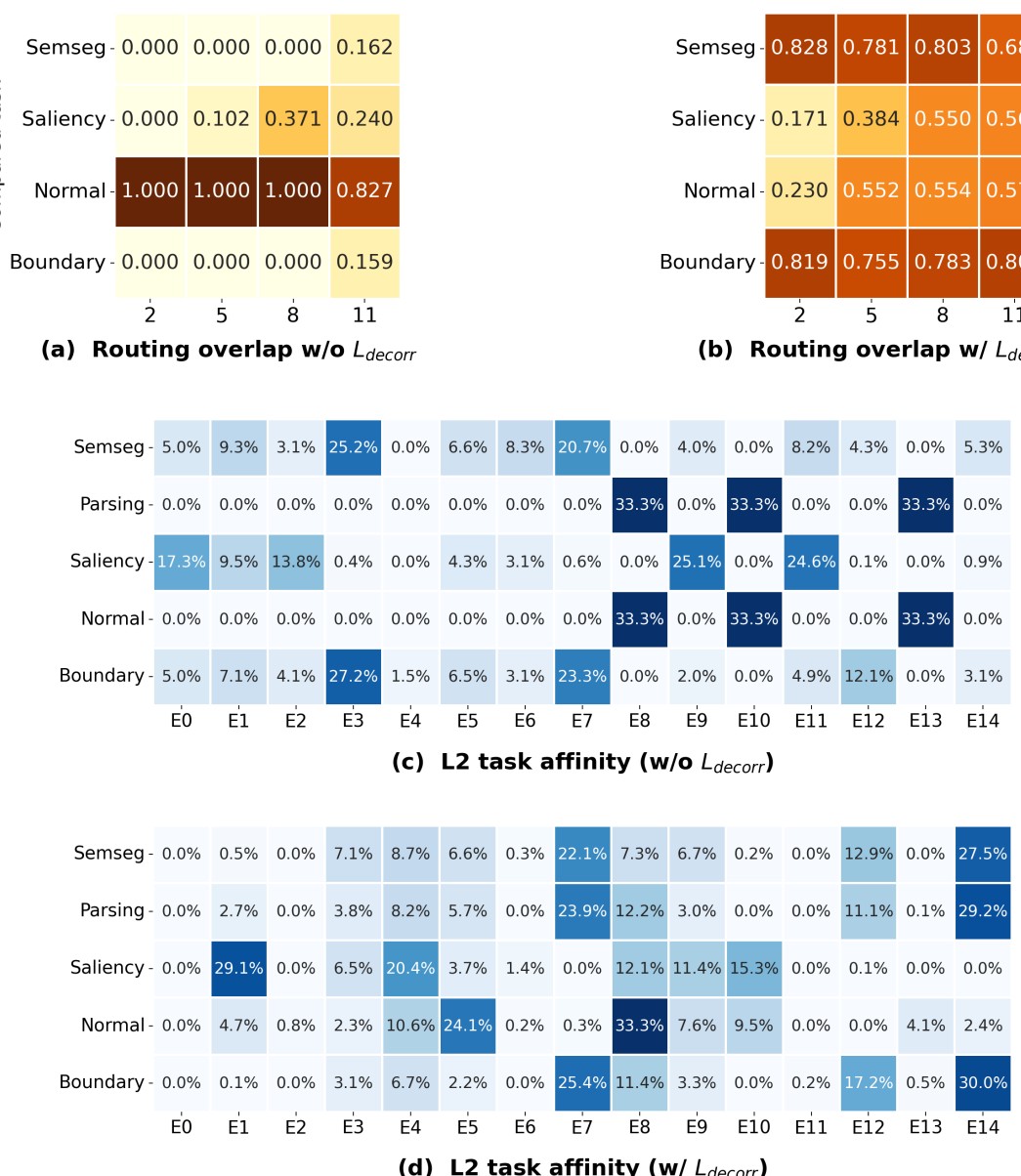

*Figure 9.* **Why Human Parsing benefits more from** $\mathcal{L}_{decorr}$. (a,b) Parsing-condition routing overlap across MoE layers. Each cell shows the token-level routing Jaccard overlap between Human Parsing and another downstream task under the corresponding task condition. Without $\mathcal{L}_{decorr}$, Parsing is almost fully aligned with Normal in early/mid layers, while its overlap with SemSeg and Boundary is nearly zero until Layer 11. With $\mathcal{L}_{decorr}$, this pattern changes substantially: Parsing–Normal overlap decreases, while overlap with SemSeg and Boundary increases markedly. (c,d) Layer-2 task affinity, i.e., expert usage distribution. Without $\mathcal{L}_{decorr}$, Parsing and Normal concentrate on nearly the same sparse experts; with $\mathcal{L}_{decorr}$, Parsing redistributes toward experts that are also used by SemSeg/Boundary. Together, these results suggest that $\mathcal{L}_{decorr}$ reorganizes Parsing away from an overly geometry-dominated routing pattern toward a more task-appropriate combination of semantic and boundary-sensitive sharing.

Figure 9 further explains the routing mechanism behind this gain. Without $\mathcal{L}_{decorr}$, Human Parsing is almost fully aligned with Normal in early and middle layers, and both tasks concentrate on nearly the same sparse experts. With $\mathcal{L}_{decorr}$, Parsing–Normal overlap decreases substantially, while overlap with SemSeg and Boundary increases. This suggests that $\mathcal{L}_{decorr}$ does not merely enforce stronger global orthogonality; it reorganizes Parsing toward a task-appropriate mixture of semantic and boundary-sensitive sharing.

Table 7 reports overall Human Parsing and boundary-band metrics. The gain from $\mathcal{L}_{decorr}$ is not limited to global mIoU; it

remains clear near part boundaries, where boundary-band mIoU improves by $+5.59$ and foreground-only boundary-band mIoU improves by $+5.86$.

Table 7. **Overall Human Parsing analysis.** Boundary-band metrics are computed near ground-truth part boundaries.

| Metric | w/o $\mathcal{L}_{decorr}$ | w/ $\mathcal{L}_{decorr}$ | $\Delta$ |
|---|---|---|---|
| Human Parsing mIoU (all) | 63.69 | **71.18** | **+7.48** |
| Human Parsing mIoU (fg only) | 58.54 | **67.08** | **+8.54** |
| Boundary-band mIoU (all) | 42.26 | **47.85** | **+5.59** |
| Boundary-band mIoU (fg only) | 40.74 | **46.60** | **+5.86** |
| Boundary-band pixel acc. | 62.59 | **66.99** | **+4.40** |

The class-wise results in Table 8 show that the largest gains occur on fine-grained limb categories, such as upper arm, lower arm/hand, and lower leg/foot. These are precisely the categories where neighboring body parts are semantically similar and separated by thin local boundaries.

Table 8. **Per-class IoU on Human Parsing.**

| Class | w/o $\mathcal{L}_{decorr}$ | w/ $\mathcal{L}_{decorr}$ | $\Delta$ |
|---|---|---|---|
| Background | 94.58 | **95.72** | +1.14 |
| Head | 86.31 | **90.43** | +4.12 |
| Torso | 71.88 | **77.92** | +6.04 |
| Upper arm | 48.97 | **62.24** | **+13.28** |
| Lower arm / hand | 53.10 | **63.53** | **+10.43** |
| Upper leg | 50.91 | **57.59** | +6.67 |
| Lower leg / foot | 40.09 | **50.80** | **+10.71** |

Table 9 confirms the same trend under boundary-band evaluation. The strongest boundary-region gains again appear on articulated limb parts, supporting that $\mathcal{L}_{decorr}$ improves local part separation rather than only coarse semantic recognition.

Table 9. **Boundary-band per-class IoU on Human Parsing.**

| Class | w/o $\mathcal{L}_{decorr}$ | w/ $\mathcal{L}_{decorr}$ | $\Delta$ |
|---|---|---|---|
| Background | 51.36 | **55.37** | +4.01 |
| Head | 55.95 | **60.46** | +4.52 |
| Torso | 44.89 | **49.21** | +4.32 |
| Upper arm | 34.10 | **40.83** | **+6.72** |
| Lower arm / hand | 41.02 | **45.60** | +4.58 |
| Upper leg | 34.78 | **41.14** | **+6.36** |
| Lower leg / foot | 33.72 | **42.35** | **+8.63** |

## D. Sensitivity Analysis

We evaluate the sensitivity of PRISM to routing sparsity, expert capacity, the shared expert, and the application depth of $\mathcal{L}_{decorr}$ on PASCAL-Context with a ViT-S student. As shown in Table 10, nearby configurations remain competitive, indicating that PRISM is not a brittle single-point design. The default setting provides the best overall balance, especially on Human Parsing and Normal Estimation.

The routing sparsity results show that Top-1 underuses expert combinations, while Top-4 increases capacity but hurts Boundary, suggesting that overly dense routing weakens specialization. Varying the number of experts from 12 to 18 produces stable but slightly lower results than the default $N = 15$. Removing the shared expert also degrades performance, confirming that a small shared component inside the sparse path stabilizes routing. Finally, applying $\mathcal{L}_{decorr}$ across L2,5 gives the best multi-task trade-off, whereas removing it severely hurts Human Parsing.

*Table 10.* **Sensitivity analysis on PASCAL-Context with ViT-S.**

| Configuration | Semseg mIoU ↑ | Parsing mIoU ↑ | Saliency maxF ↑ | Normal mErr ↓ | Boundary odsF ↑ |
|---|---|---|---|---|---|
| $N = 15$, Top-1 | 77.16 | 66.77 | 84.43 | 14.79 | 68.47 |
| $N = 15$, Top-2 | 78.97 | 68.91 | 84.68 | 14.66 | 69.02 |
| $N = 15$, Top-4 | 78.81 | 68.92 | 84.45 | 14.62 | 66.43 |
| $N = 12$, Top-3 | 78.21 | 68.56 | 84.89 | 14.78 | 68.84 |
| $N = 18$, Top-3 | 78.34 | 68.68 | 84.37 | 14.70 | 69.01 |
| w/o Shared Expert | 78.72 | 67.73 | 84.20 | 14.88 | 68.01 |
| w/o $\mathcal{L}_{decorr}$ | **79.80** | 61.80 | **85.50** | 14.32 | **70.86** |
| $\mathcal{L}_{decorr}$ on L2 | 79.12 | 68.96 | 84.53 | 14.51 | 68.27 |
| $\mathcal{L}_{decorr}$ on L2,5,8 | 78.80 | 68.32 | 84.37 | 14.45 | 68.24 |
| PRISM Default | 79.19 | **69.25** | 85.01 | **14.28** | 70.78 |

## E. Reduced-Teacher Distillation

We further test whether PRISM still helps when fewer teachers are distilled. Table 11 reports a reduced-teacher setting using only DINOv2 and SAM. Under the same student and training setup, PRISM clearly outperforms a dense widened student. This indicates that PRISM's benefit does not rely on the full CLIP+DINOv2+SAM teacher set; even with fewer heterogeneous teachers, sparse context-dependent routing remains useful for separating and recombining complementary knowledge.

*Table 11.* **Reduced-teacher setting with DINOv2 + SAM on PASCAL-Context.**

| Model | Semseg mIoU ↑ | Parsing mIoU ↑ | Saliency maxF ↑ | Normal mErr ↓ | Boundary odsF ↑ |
|---|---|---|---|---|---|
| Wide ViT | 66.13 | 61.21 | 84.16 | 14.73 | 67.08 |
| PRISM | **77.80** | **68.90** | **84.50** | **14.43** | **69.48** |

## F. Additional Efficiency Details

PRISM introduces additional offline training cost because Stage 1 requires forwarding frozen VFM teachers. This cost is paid only during distillation; after training, inference uses a single student model without running multiple VFMs. Table 12 summarizes the end-to-end training cost for the main ViT-B PASCAL setting.

*Table 12.* **End-to-end training cost of PRISM on PASCAL-Context with ViT-B.** GPU-hours include teacher forwarding.

| Stage | Schedule | Train / Val Batch size | Hardware | GPU-hours |
|---|---|---|---|---|
| Stage 1 | 30 epochs | 20 / 20 | RTX 5090 32GB | 499.2 |
| Stage 2 | 40k iters | 2 / 8 | RTX 5090 32GB | 56.1 |
| **Total** | – | – | – | **555.3** |

## G. Additional Analysis of Routing Dynamics

We present a detailed, layer-by-layer visualization of the routing dynamics. In the following figures, each heatmap spans the full page width to reveal fine-grained activation patterns.

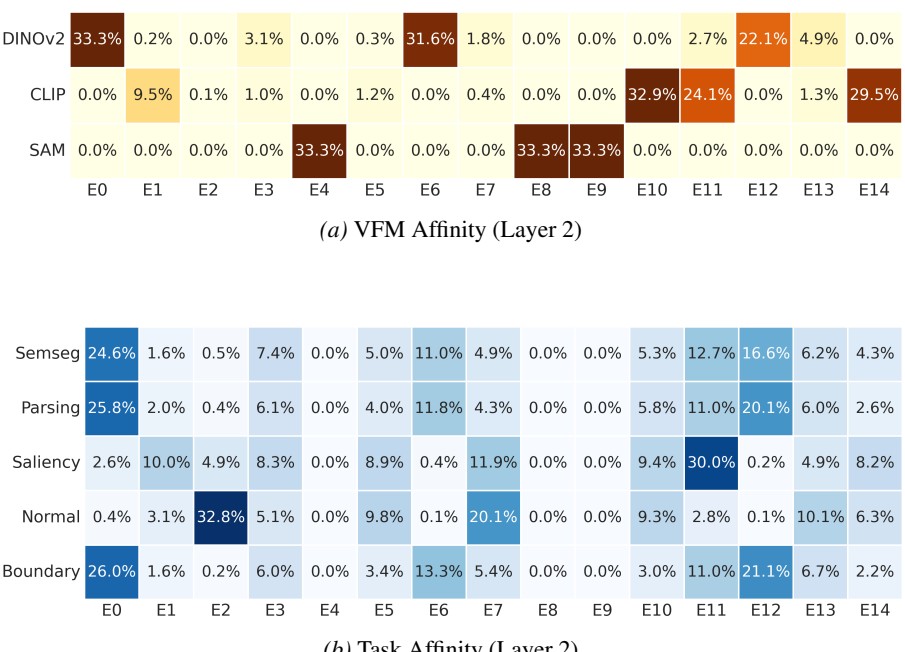

*(a)* VFM Affinity (Layer 2)

*(b)* Task Affinity (Layer 2)

*Figure 10.* **Routing Dynamics at Shallow Layer (Layer 2).** *Top:* VFM Affinity. *Bottom:* Task Affinity. At this early stage, the activation blocks are broad and diffuse. The router does not yet distinguish sharply between teachers or tasks, indicating that Layer 2 processes universal, shared visual primitives (Global Consensus).

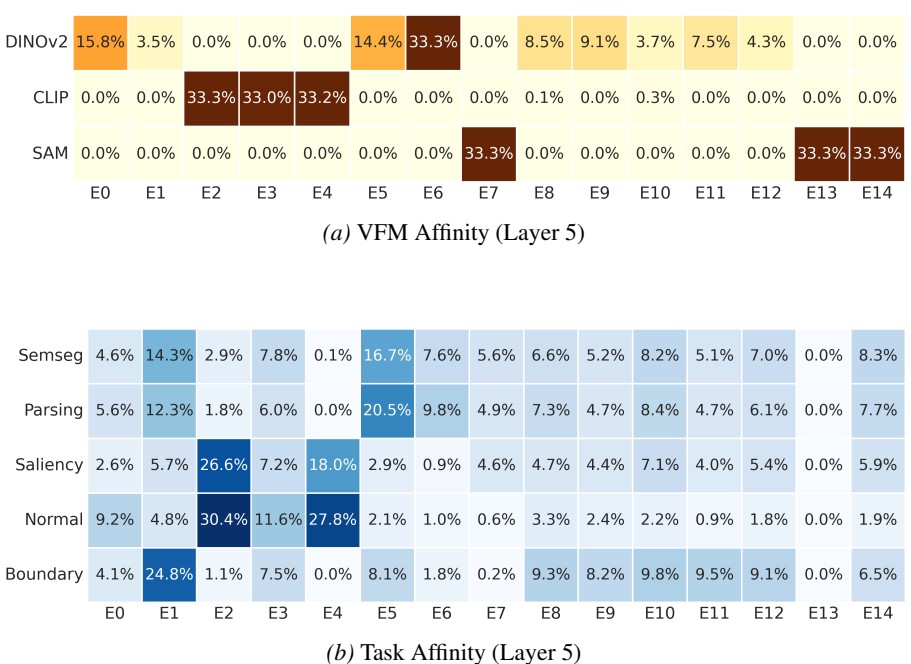

*(a)* VFM Affinity (Layer 5)

*(b)* Task Affinity (Layer 5)

*Figure 11.* **Routing Dynamics at Middle Layer (Layer 5).** Transitioning to mid-level features, we observe the onset of separation. While some overlaps persist, the router begins to form distinct clusters for different contexts, preparing for the semantic split in deeper layers.

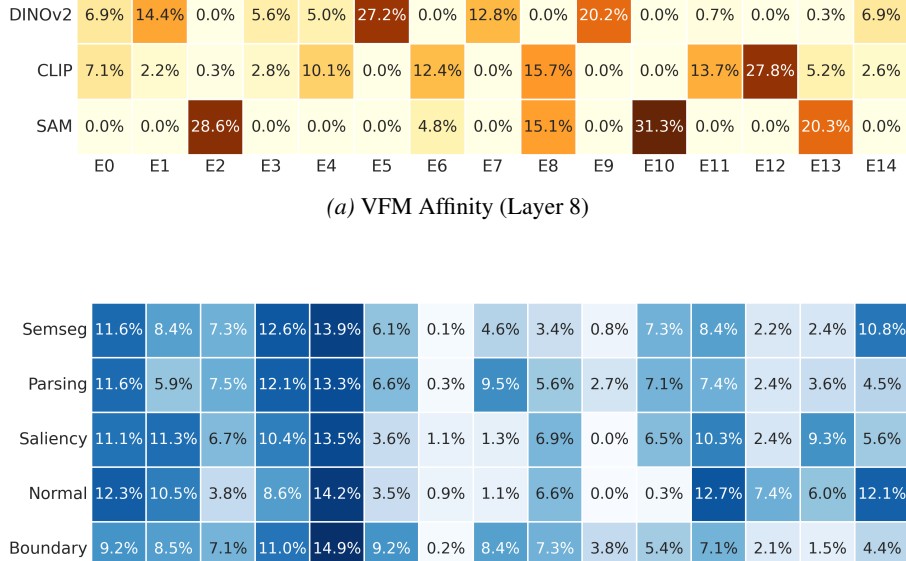

*(a)* VFM Affinity (Layer 8)

*(b)* Task Affinity (Layer 8)

*Figure 12.* **Routing Dynamics at Deep Layer (Layer 8).** In the deep layers, the experts exhibit sharp, context-dependent specialization resembling Layer 11. Distinct, non-overlapping clusters form for each teacher (Top), supporting that effective conflict reduction is mainly handled in the network's later stages.

