# OpenReview forum: "PRISM: Synergizing Vision Foundation Models via Self-organized Expert Specialization"
_ICML.cc/2026/Conference — ICML 2026 regular_

### Official Review · Reviewer_Bwop · 2026-03-09

**Soundness:** 3
**Presentation:** 3
**Significance:** 3
**Originality:** 3
**Overall Recommendation:** 4
**Confidence:** 4

**Summary:**

This paper proposes PRISM, addressing the feature conflict challenge in multi-teacher distillation by introducing a dual-stream MoE framework designed to synergize diverse Vision Foundation Models through a "deconstruction-then-recombination" paradigm. The architecture balances stability and plasticity by combining a Universal Anchor stream with a Conditioned MoE stream, utilizing a FiLM-based teacher-conditional router to guide experts toward specialized representational subspaces during an initial expertise deconstruction stage. To prevent redundant expert states and promote feature diversity, the authors propose a local-aware decoupling loss that preserves high-rank representations at the local level. By effectively isolating conflicting gradients during training and dynamically recombining specialized skills for downstream tasks, PRISM achieves state-of-the-art performance on dense prediction benchmarks such as PASCAL-Context and NYUD-v2 while maintaining computational efficiency.

**Compliance With Llm Reviewing Policy:**

Affirmed.

**Key Questions For Authors:**

See weaknesses above.

**Limitations:**

yes

**Strengths And Weaknesses:**

### Strengths
- The paper provides a principled approach to the problem of negative transfer in multi-teacher distillation. The identification of gradient conflict as a primary bottleneck is well-supported by both theoretical motivation and empirical evidence.
- The experiments are conducted on established and challenging multi-task benchmarks (PASCAL-Context, NYUD-v2). Comparing the PRISM encoder against highly specialized teachers (CLIP, DINOv2, SAM) and contemporary unified models (SAK, RADIOv2.5) ensures the claims of teacher synergy are verified against the current state-of-the-art.
- The distinction between the Universal Anchor (stability) and MoE (plasticity) streams is an intuitive way to describe the architecture.

### Weaknesses
- The performance of $\mathcal{L}_{decorr}$ and the FiLM modulation likely depends on careful balancing of loss weights. A more thorough sensitivity analysis of the proposed hyperparameters (e.g., $\mu, \alpha, \beta$) would strengthen the claim of the method's robustness.
- Lacking related works about the application of FiLM in MoE. What are the main differences between PRISM with MoFME (R. Zhang et al, Efficient Deweather Mixture-of-Experts with Uncertainty-aware Feature-wse Linear Modulation, AAAI 2024)?
- The author should discuss the efforts made to achieve fair comparison, especially in the main Tables, since PRISM uses a dual-stream strategy.
- What is the purpose of the comparison in Table 4? The conclusion of this version seems to be the superiority of MoE, but not the proposed dynamic routing. If the author wants to show the performance gain does not come from the increased active parameters by the proposed approaches (anchor + shared experts + top-3 experts), you should compare with a vanilla MoE (with similar parameter counts) rather than a dense vit. Or, is this “Wide ViT-S” exactly an MoE? Then, what is the routing scheme of it?

---

> ### Author Rebuttal · Authors · 2026-03-31
>
> We sincerely thank the reviewer for the meticulous review, highly constructive feedback, and insightful suggestions.
>
> ### w1. Sensitivity / robustness of FiLM modulation and routing
> We agree that robustness is important for a routing-based method. To address this, we added a broader sensitivity analysis on **PASCAL-Context with a ViT-S student**, varying **Top-k**, expert capacity **N**, the shared expert, and the application depth of **$L_{decorr}$**. The results show that PRISM is not a brittle single-point design: performance remains consistently strong across a range of reasonable settings, while the default setting provides the best overall multi-task trade-off.
>
> | Configuration | Semseg ↑ | Parsing ↑ | Saliency ↑ | Normals ↓ | Boundary odsF ↑ |
> |-|-:|-:|-:|-:|-:|
> | (1) N=15, Top-1 | 77.16 | 66.77 | 84.43 | 14.79 | 68.47 |
> | (2) N=15, Top-2 | 78.97 | 68.91 | 84.68 | 14.66 | 69.02 |
> | (3) N=15, Top-4 | 78.81 | 68.92 | 84.45 | 14.62 | 66.43 |
> | (4) N=12, Top-3 | 78.21 | 68.56 | 84.89 | 14.78 | 68.84 |
> | (5) N=18, Top-3 | 78.34 | 68.68 | 84.37 | 14.70 | 69.01 |
> | (6) w/o Shared Expert | 78.72 | 67.73 | 84.20 | 14.88 | 68.01 |
> | (7) w/o L_decorr | **79.80** | 61.80 | **85.50** | 14.32 | **70.86** |
> | (8) L_decorr on L2 | 79.12 | 68.96 | 84.53 | 14.51 | 68.27 |
> | (9) L_decorr on L2,5,8 | 78.80 | 68.32 | 84.37 | 14.45 | 68.24 |
> | **(10) PRISM Default (Top-3, L2,5,8,11)** | 79.19 | **69.25** | 85.01 | **14.28** | 70.78 |
>
> We also clarify that $\gamma(c)$, $\beta(c)$, and $\lambda$ are **learned internal modulation variables**, not manually tuned hyperparameters. Specifically, $\gamma(c)$ and $\beta(c)$ are FiLM-generated affine coefficients, and their prediction head is **zero-initialized**, so conditioning starts from an identity mapping and is learned progressively rather than imposed a priori. Likewise,$\lambda$ is a learned gate that adaptively balances the Universal Anchor and Specialized Experts. Empirically, $\lambda$ shows a consistent depth-dependent convergence pattern, around **0.7 in shallow layers** and around **0.5 in deeper layers**, suggesting a stable emergent balance rather than a fragile hand-tuned trade-off. We will add this clarification to make it clear that the method does not rely on delicate manual tuning of these internal variables.
>
> ### w2. FiLM in MoE and relation to prior FiLM-based MoE work
> Thank you for highlighting this related work. We agree that MoFME should be discussed in the related-work section, and we will add it and other prior FiLM-in-MoE papers in the revision.
> PRISM differs in both objective and usage of FiLM: MoFME studies efficient all-in-one deweathering, where modulation is used to realize lightweight experts, while in PRISM FiLM is only used to modulate the router under different teacher/task conditions in an explicit sparse MoE. Accordingly, PRISM’s contribution is centered on resolving cross-teacher interference via dual-stream decomposition and recomposition, rather than introducing FiLM into MoE per se.
>
> ### w3. Fair comparison under a dual-stream design
> We agree to explicitly discuss fairness regarding our dual-stream block. Table 4 compares PRISM to a **dense Wide ViT-S** (FFN width scaled 5x via Net2Net) under **matched inference FLOPs**. This Wide ViT-S is **not** an MoE; it controls for **static dense capacity scaling**, not routing.
>
> Following your suggestion, we added a strict vanilla MoE baseline under the same full-teacher setting, stripping all PRISM-specific components (Anchor, FiLM routing, shared expert, dual-stream) to leave only plain sparse MoE routing, with top-5 out of 15 experts selected to match the capacity of PRISM.
>
> | Model | Semseg ↑ | Parsing ↑ | Saliency ↑ | Normals ↓ | Boundary odsF ↑ |
> |-|-:|-:|-:|-:|-:|
> | vanilla MoE | 72.96 | 61.77 | 83.34 | 15.03 | 66.73 |
> | dense Wide ViT-S | 78.15 | 67.71 | 84.88 | **14.24** | 70.35 |
> | **PRISM Default** | **79.19** | **69.25** | **85.01** | 14.28 | **70.78** |
>
> These results confirm the performance gain is not explained solely by **static capacity scaling** (as PRISM outperforms dense Wide ViT-S), nor by merely “using any MoE” (as PRISM substantially outperforms vanilla MoE). We will make this explicit in the revision.
>
> ### w4. Purpose of Table 4
> The purpose of Table 4 is **not** to prove the superiority of MoE in general. Rather, it isolates a narrower question: under **matched inference FLOPs**, is **dynamic routing** more effective than **static dense scaling**? In that sense, Wide ViT-S is a dense static-capacity baseline, not an MoE and not a routing baseline. We agree that Table 4 by itself does not isolate the benefit of PRISM-specific routing from sparse MoE capacity; this is exactly why the additional **vanilla MoE** comparison above is important. We will revise the wording accordingly and present the two controls more clearly: **Wide ViT-S** for static dense scaling, and **vanilla MoE** for plain sparse routing.

---

> > ### Author Rebuttal · Reviewer_Bwop · 2026-04-01
> >
> > I thank the authors for their rebuttal to address all my concerns. I would like to keep my positive score. Good luck.

---

> > > ### Author Response · Authors · 2026-04-02
> > >
> > > Thank you for your encouraging feedback. We are very glad that our response has adequately addressed your concerns. We will make sure the final revision reflects the clarified positioning and improved presentation discussed in the rebuttal. We sincerely appreciate your time and support.

---

### Official Review · Reviewer_siEV · 2026-03-11

**Soundness:** 3
**Presentation:** 3
**Significance:** 3
**Originality:** 3
**Overall Recommendation:** 5
**Confidence:** 4

**Summary:**

To overcome the gradient conflicts inherent in distilling heterogeneous Vision Foundation Models (VFMs) into a single network, this paper introduces PRISM, a novel dual-stream Mixture-of-Experts (MoE) framework. Instead of relying on rigid, static parameter partitioning, PRISM dynamically routes features through a Universal Anchor stream for shared consensus and a Conditioned MoE stream to resolve conflicting representational demands. The model operates on a two-stage "Decompose-then-Recombine" paradigm: first, a context-modulated router utilizes teacher IDs alongside a locality-aware decorrelation loss to guide experts into specializing in distinct visual modalities; then, it dynamically recombines these learned primitives for specific downstream applications. Extensive experiments confirm that PRISM sets a new state-of-the-art on the PASCAL-Context and NYUD-v2 benchmarks, successfully synergizing diverse visual knowledge domains with high parameter efficiency.

**Compliance With Llm Reviewing Policy:**

Affirmed.

**Final Justification:**

The rebuttal and follow-up have fully addressed my concerns. Therefore, I would like to raise my rating to "5: Accept".

**Key Questions For Authors:**

Please see the Weaknesses section above.

**Limitations:**

The limitations could be discussed regarding:
- Under-explored scalability
- Performance trade-offs on high-frequency tasks (e.g., NYUD-v2 Boundary Detection)
- Two-stage training overhead

**Strengths And Weaknesses:**

## Strengths
1. The paper successfully reframes multi-teacher distillation by moving away from static, manual "hard boundary" partitioning. Instead, PRISM leverages dynamic, self-organized emergence to handle the soft boundaries and intricate overlaps inherent in visual knowledge.

2. PRISM establishes a new state-of-the-art on the PASCAL-Context benchmark, outperforming prior methods like SAK and RADIOv2.5. It also demonstrates strong competitiveness on the NYUD-v2 indoor scene benchmark.

3. The dual-stream mechanism elegantly balances optimization needs. The Universal Anchor stream maintains stability for shared consensus, while the Conditioned MoE stream provides the plasticity needed to resolve conflicting requirements.

4. The authors provide compelling empirical validation of their theory regarding gradient orthogonalization. They demonstrate through cosine similarity and topological distributions that their sparse experts successfully decouple conflicting tasks to nullify interference, whereas standard dense models suffer from gradient averaging.

## Weaknesses
1. The framework relies on a custom "Locality-Aware Decorrelation Loss" to ensure input tokens remain diverse, preventing the router from lacking discriminative signals. However, the paper does not justify why this computationally complex, spatial-based penalty is necessary compared to much simpler, standard MoE techniques like expert load-balancing regularization or routing entropy penalties. The lack of an ablation comparing their custom loss to these established baselines makes it difficult to assess if the added complexity is truly necessary.

2. In Section 4.3 and the Figure 2 caption, the authors claim that the shared FFN parameters (represented by red diamonds) "cluster along the diagonal y=x," which they use as primary visual evidence of gradient conflict. However, Figure 2(a) actually shows that these diamonds frequently appear off-diagonal (e.g., near coordinates (0.45, 1.0) and (1.0, 0.35) in the DINOv2 vs. CLIP plot, and near (0.1, 1.0) in the DINOv2 vs. SAM plot).

3. While highly effective overall, PRISM slightly underperforms the SAK baseline in Boundary Detection on the NYUD-v2 dataset. The authors attribute this to SAK's use of dedicated, physically separated adapters that are specifically tuned for high-frequency cues.

4. The core architectural implementation and primary ablation studies are focused on ViT-B/16 and ViT-S student backbones. The paper does not explicitly explore how this emergent specialization scales to larger student architectures.

---

> ### Author Rebuttal · Authors · 2026-03-31
>
> We thank the reviewer for the positive assessment and for the very concrete suggestions on where the original claims should be made more precise.
>
> ### w1. Why is $L_{decorr}$ needed beyond standard MoE regularizers?
> We agree this comparison is essential. We clarify that our original ablation already includes the standard **load-balancing loss**. Therefore, the comparison is not “regularized vs. unregularized,” but rather **load-balancing only vs. load-balancing +  $L_{decorr}$**.
>
> On **PASCAL-Context with a ViT-S student**, we compare the original w/o $L_{decorr}$ setting against the default PRISM configuration. This isolates the effect of $L_{decorr}$ under the same student scale and task setup:
>
> | Model / Reg. (ViT-S, PASCAL-Context) | Semseg ↑ | Parsing ↑ | Saliency ↑ | Normals ↓ | Boundary odsF ↑ |
> |---|---:|---:|---:|---:|---:|
> | load-balancing only (w/o $L_{decorr}$) | **79.80** | 61.80 | **85.50** | 14.32 | **70.86** |
> | **PRISM Default (Top-3, L2,5,8,11)** | 79.19 | **69.25** | 85.01 | **14.28** | 70.78 |
>
> While some metrics remain strong without  $L_{decorr}$, Human Parsing drops sharply from **69.25 to 61.80**, indicating that standard balancing alone does not prevent collapse toward an imbalanced solution. This is exactly why we view  $L_{decorr}$ as a mechanism for **balanced multi-task specialization**, rather than a regularizer that must improve every metric independently.
>
> We further tested a stronger **routing-entropy baseline** under the **main ViT-B setting**. Here, both models use  $L_{decorr}$, and we compare adding an entropy penalty to the router against the full PRISM training recipe:
>
> | Model / Reg. (ViT-B, main setting) | Semseg ↑ | Parsing ↑ | Saliency ↑ | Normals ↓ | Edge loss ↓ |
> |---|---:|---:|---:|---:|---:|
> | routing entropy penalty + $L_{decorr}$ | 79.81 | 68.87 | 84.54 | **13.76** | 1.293 |
> | **PRISM Default** | **82.36** | **74.82** | **85.34** | 13.77 | **1.244** |
>
> This comparison serves a different purpose: it shows that even after adding a stronger standard router regularizer, the result still remains clearly below the full method, especially on semseg, parsing, saliency, and edge quality. Taken together, these two experiments support a narrower and more precise claim: **simpler MoE regularizers help, but they do not replace $L_{decorr}$ in our setting**.
>
> ### w2. Figure 2 wording about shared FFN “clustering along the diagonal”
> We thank you for observing Fig 2(a) closely. We agree the shared FFN parameters (red diamonds) are not strictly confined to $y=x$. "Clustering along the diagonal" describes a **macro-statistical trend of simultaneous activation**, not equal gradient magnitudes.
>
> The evidence of conflict is the distinction between the diagonal zone and the axes. Off-diagonal points like (0.45, 1.0) show a parameter receiving strong, non-zero updates from *both* VFMs simultaneously, constituting significant gradient interference.
>
> In contrast, MoE experts (blue circles) strictly hug the axes ($x \approx 0$ or $y \approx 0$), proving PRISM successfully decouples these updates. We will revise the caption to state the FFN exhibits a "broad correlation of simultaneous updates" rather than strict alignment.
>
> ### w3. Boundary Detection underperformance
> Boundary prediction requires high-frequency local cues; semantic tasks need global abstraction. SAK and MLoRE excel here as isolated adapters preserve local signals. Under limited capacity (ViT-S/B), PRISM prioritizes semantic disentanglement. Removing $\mathcal{L}_{\text{decorr}}$ slightly improves boundary (70.78→70.86) but severely hurts parsing (69.25→61.80), showing a multi-task trade-off. This gap stems from capacity limits, not mechanism: our ViT-L (10% data) achieves better boundary (76.36 vs SAK's 76.27) and Normal (13.49 vs 13.82), supportting that PRISM can captures both global and local cues  given sufficient capacity. We will complete the full ViT-L experiments, and include in the revision.
>
> ### w4. Scaling to larger students
>
> Thank you for your suggestion. We added a **preliminary ViT-L** experiment with **Stage 1: ImageNet 10% subset pretraining** and **Stage 2: PASCAL-Context fine-tuning**. Due to the rebuttal time constraint, we only provide **reduced-budget** result, this is not a direct comparison to full-budget ViT-L results in prior work; instead, we use it as **initial evidence that PRISM remains stable when scaled to ViT-L**. We will finish the full scale experiment after the rebuttal and include in the paper.
>
> | Configuration | Semseg ↑ | Parsing ↑ | Saliency ↑ | Normals ↓ | Boundary odsF ↑ |
> |---|---:|---:|---:|---:|---:|
> | Single-task baseline ViT-L | 81.61 | 72.77 | 83.80 | 13.87 | 75.24 |
> | SAK ViT-L *(ImageNet-1K → PASCAL-Context)* | **84.01** | **76.99** | 84.65 | 13.82 | 76.27 |
> | **PRISM ViT-L *(ImageNet-1K 10% subset → PASCAL-Context)*** | 82.84 | 76.72 | **84.74** | **13.49** | **76.36** |

---

> > ### Author Rebuttal · Reviewer_siEV · 2026-03-31
> >
> > The authors' response is greatly appreciated. Most of my previous concerns have been addressed. I just have a minor remaining concern about the comparison between "load-balancing only" and "load-balancing + $L_{decorr}$", as the performance metrics on most benchmarks are closely comparable, except "Human Parsing". It is still unclear how $L_{decorr}$ would benefit human parsing specifically, or this difference is just a random noise. Could authors provide more insights on why this task's performance comparison is special?

---

> > > ### Author Response · Authors · 2026-04-06
> > >
> > > Thank you for this thoughtful question. We appreciate the reviewer highlighting this point. To assess whether the stronger improvement on Human Parsing is simply random noise or instead reflects a task-specific benefit of  $L_{decorr}$, we conducted three additional analyses: **(1)** multi-seed reruns of the w/o $L_{decorr}$ variant, **(2)** a dedicated Human Parsing error analysis including per-class IoU and boundary-band evaluation, and **(3)** a routing / expert-specialization analysis on the Parsing task. **Taken together, these results strongly suggest that the stronger gain on Human Parsing is not random noise, but a genuine task-specific benefit of $L_{decorr}$.**
> > >
> > >
> > > ### 1. Multi-seed reruns of the w/o $L_{decorr}$ variant
> > >
> > > We reran the **w/o $L_{decorr}$** variant with three seeds. Overall, performance is fairly stable across runs. Parsing shows some run-to-run variation (**61.80–62.14 mIoU**), but this is comparable in scale to Semseg and does not by itself support a strong claim of unusual seed sensitivity. We therefore avoid overstating the task-level effect of $L_{decorr}$ from a single run, and instead focus on more diagnostic analyses of **where** the gain appears.
> > > | Setting | Seed | Semseg ↑ | Parsing ↑ | Saliency ↑ | Normals ↓ | Boundary loss ↓ |
> > > |--|--|--:|--:|--:|--:|--:|
> > > | w/o \(L_{decorr}\) | seed_32 | 79.61 | 62.14 | 85.46 | 14.26 | 1.38 |
> > > | w/o \(L_{decorr}\) | seed_42 | 79.80 | 61.80 | 85.50 | 14.32 | 1.40 |
> > > | w/o \(L_{decorr}\) | seed_52 | 79.48 | 61.85 | 85.49 | 14.26 | 1.41 |
> > >
> > > ### 2. Human Parsing per-class and boundary-band analysis
> > >
> > > To better understand the effect of $L_{decorr}$ on Human Parsing, we analyze **per-class IoU** and **boundary-band performance**. The improvement is highly non-uniform across categories, with the largest gains  on upper arm (+13.28), lower arm/hand (+10.43), and lower leg/foot (+10.71). The same trend persists under **boundary-band evaluation**, which measures performance only within a narrow band around ground-truth part boundaries: the largest gains appear on upper arm (+6.72), upper leg (+6.36), and lower leg/foot (+8.63). These are precisely the categories where adjacent parts are semantically similar and separated by thin, ambiguous boundaries. This trend is also visible qualitatively in **[Fig. E](https://anonymous.4open.science/r/anonymous_rebuttal/anonymous_rebuttal_2.md)**: across three representative examples, $L_{decorr}$ produces more coherent lower-body and limb regions, fewer fragmented predictions, and in one case a part layout closer to the annotation.
> > >
> > > **Per-class IoU on Human Parsing**
> > >
> > > | Class | w/o $L_{decorr}$ | w/ $L_{decorr}$ | Delta |
> > > |-|-:|-:|-:|
> > > | Background | 94.58 | **95.72** | +1.14 |
> > > | Head | 86.31 | **90.43** | +4.12 |
> > > | Torso | 71.88 | **77.92** | +6.04 |
> > > | Upper arm | 48.97 | **62.24** | **+13.28** |
> > > | Lower arm / hand | 53.10 | **63.53** | **+10.43** |
> > > | Upper leg | 50.91 | **57.59** | +6.67 |
> > > | Lower leg / foot | 40.09 | **50.80** | **+10.71** |
> > >
> > > **Boundary-band per-class IoU on Human Parsing**
> > >
> > > | Class | w/o $L_{decorr}$ | w/ $L_{decorr}$ | Delta |
> > > |-|-:|-:|-:|
> > > | Background | 51.33 | **55.10** | +3.77 |
> > > | Head | 43.70 | **47.23** | +3.53 |
> > > | Torso | 39.93 | **40.32** | +0.39 |
> > > | Upper arm | 34.10 | **40.83** | **+6.72** |
> > > | Lower arm / hand | 41.02 | **45.60** | **+4.58** |
> > > | Upper leg | 34.78 | **41.14** | **+6.36** |
> > > | Lower leg / foot | 33.72 | **42.35** | **+8.63** |
> > >
> > > ### 3. Task-conditioned routing and expert usage for Human Parsing
> > >
> > > To understand why Parsing benefits most, we analyze task-conditioned routing and expert usage in **[Fig. E](https://anonymous.4open.science/r/anonymous_rebuttal/anonymous_rebuttal_2.md)**. **Without $L_{decorr}$ ([Fig. E(a,c)](https://anonymous.4open.science/r/anonymous_rebuttal/anonymous_rebuttal_2.md))**, Parsing is strongly entangled with Normals: their routing overlap is nearly 1.0 in layers 2/5/8, and both tasks concentrate on almost the same sparse experts. **With $L_{decorr}$ ([Fig. E(b,d)](https://anonymous.4open.science/r/anonymous_rebuttal/anonymous_rebuttal_2.md))**, Parsing–Normals overlap decreases substantially (e.g., 1.000 → 0.230  at L2), while overlap with Semseg and Boundary increases markedly (e.g., Parsing–Semseg: 0 → 0.828, Parsing–Boundary: 0 → 0.819 at L2). This suggests that $L_{decorr}$ does not simply enforce stronger global orthogonality; it **reorganizes** Parsing away from an overly geometry-dominated routing pattern toward a more task-appropriate combination of semantic and boundary-sensitive sharing. We believe this is why the effect of $L_{decorr}$ is especially visible on Human Parsing.

---

### Official Review · Reviewer_Pt7x · 2026-03-12

**Soundness:** 2
**Presentation:** 2
**Significance:** 2
**Originality:** 3
**Overall Recommendation:** 4
**Confidence:** 3

**Summary:**

The paper proposes PRISM, a framework for combining knowledge from multiple vision foundation models, specifically using CLIP, DINO-v2, and SAM into a single small student model . When you try to learn from multiple teachers simultaneously, they often give contradict  signals which reduce the overall performance -- this is the core problem the paper tries to solve. At a high level, the approach uses a mixture of experts architecture where different experts are sub-networks automatically specialized in different types of visual knowledge. The routing mechanism decides which expert handles which input. This approach is divided into two stages: (1) decompose stage: trains a student to mimic each teacher with a teacher conditioned router that guides the expert to specialize. (2) recombine stage: there is a fine-tuning that happens for the downstream task where the router learns to combine experts appropriately per task. Additionally, the paper also introduces a locality-aware decorrelation loss

**Compliance With Llm Reviewing Policy:**

Affirmed.

**Final Justification:**

Most of my concerns were addressed, so I am updating the score to reflect the same

**Key Questions For Authors:**

The difference in performance between SAK and PRISM seems quite small, how do they compare in terms of cost/compute/time?  i.e., (1) What is the actual training cost of PRISM compared to SAK and the baseline? (2) Is the performance gain worth the added training overhead for a someone who might be interested in using PRISM? (3) Does the emergent specialization pattern remain stable across different random seeds, or is it a one-time observation?

**Limitations:**

Impact Statement is so generic and can be used for any ML paper

**Strengths And Weaknesses:**

- [S1] In terms of insight and motivation Fujgure-2a provides a lot of value, ideally this should be placed earlier in the paper

- [S2] The learnable gate $\lambda$ naturally evolving from 0.7 to 0.5 without explicit supervision is a compelling emergent behavior that validates the core idea/direction.

- [S3] Experiments are very well desgined, espcecially the Table-3 and Figures add a lot of value to the paper

- [W1] More on the paper writing side, the introduction dives almost immediately into technical nuances without establishing why this problem matters in practice. Anyone unfamiliar with VFM distillation would struggle to understand the why this problem is important. There is no compelling real-world scenario offered either in the intro or in the experiments.

- [W2] Multi-teacher distillation has a much longer history outside the VFM context. It is unclear whether the gradient conflict problem PRISM solves is already addressed in pre-LLM/VLM era literature, some discusson on this in the related work witll help

- [W3]  Paper never clearly explains what becomes newly possible because of PRISM.

- [W4] While there is some discussion on inference cost, the training cost of PRISM is never discussed. Multi-stage training with MoE typically involves significantly higher training compute.

- [W5] Sensitivity Analysis: For a method whose like PRISM, it's quite important to run some sensitivty analysis, it is genuinely unclear whether the routing patterns are robust or artifacts of a specific configuration.

---

> ### Author Rebuttal · Authors · 2026-03-31
>
> We thank the reviewer for the helpful comments on clarity, training cost, and robustness, and for the positive feedback on Figure 2(a), the learnable gate, and the experiment design.
>
> ### w1. Practical motivation / why this problem matters
> We agree the introduction needs stronger practical framing. In revision, we will foreground Figure 2(a) and the real-world scenario: deploying multiple large VFMs is expensive, while naive dense distillation causes negative transfer. PRISM instead provides a **single deployable student** for heterogeneous knowledge consolidation with reduced gradient conflict.
>
> ### w2. Relation to pre-VFM multi-teacher distillation literature
> We agree and will expand the related-work discussion. PRISM does **not** claim that prior multi-teacher KD ignored conflict. Earlier methods mostly handle it through averaging/weighting, selection, or staged distillation at the **loss/sample** level. Our setting differs: heterogeneous dense teachers can create incompatible **token-level** supervision inside one shared student. PRISM addresses this via **token-conditional sparse routing**, separating conflicting signals into more disjoint internal paths.
>
> ### w3. What becomes newly possible because of PRISM?
> The key new capability of PRISM is **conflict-aware partial sharing**: within one backbone, the model can decide at the **layer/token level** whether heterogeneous teacher knowledge should be shared, separated, or partially reused.
>
> This differs fundamentally from manually partitioned adapters/branches, whose sharing pattern is fixed a priori. PRISM instead learns a **data-dependent routing structure** from actual cross-teacher gradient interactions, enabling a continuum between shared consensus and specialized disagreement rather than an all-or-nothing choice. This makes possible **fine-grained knowledge factorization and recombination** that hard-partition methods cannot express: the same model can preserve common structure where teachers agree while allocating disjoint computation only to tokens/layers where they conflict, making PRISM attractive for consolidating multiple large VFMs into one deployable student.
>
> ### w4 & q1/q2. Training cost and whether the gain is worth it
> We now report the absolute end-to-end training cost of PRISM for the main **ViT-B Pascal** setting:
>
> | Stage | Schedule | Train batch / Val batch | Hardware |  GPU-hours  |
> |-|-|-|-|-:|
> | Stage 1 | 30 epochs | 20 / 20 | NVIDIA GeForce RTX 5090 32GB |     499.213 |
> | Stage 2 | 40k iters | 2 / 8 | NVIDIA GeForce RTX 5090 32GB |      56.072 |
> | Total | Stage 1 + Stage 2 | – | – | **555.285** |
>
> *All GPU-hours include teacher forwarding.*
>
> For fairness, we note that **SAK also uses two-stage training**, and we follow its stage-wise schedule and optimizer setup as closely as possible. Since SAK does not report end-to-end wall-clock or GPU-hour cost, we avoid unverifiable direct training-time comparisons.
>
> We therefore report a matched **ViT-S** comparison using **PRISM-lite**, which keeps the **same PRISM framework, routing, losses, and training protocol** as full PRISM and changes only one capacity knob: the **expert MLP ratio from 4.0 to 1.0**.
>
> | Model | Total Params (M) ↓ | Active Params (M) ↓ | GFLOPs ↓ | Latency (ms) ↓ |
> |-|-:|-:|-:|-:|
> | SAK ViT-S | 26.40 | 26.40 | 53.20 | 20.25 |
> | PRISM-lite ViT-S | 48.05 | 38.06 | 57.51 | 47.51 |
>
> | Model | Semseg ↑ | Parsing ↑ | Saliency ↑ | Normal ↓ | Boundary odsF ↑ | Delta_m ↑ |
> |-|-:|-:|-:|-:|-:|-:|
> | SAK ViT-S | 78.66 | 68.46 | 84.66 | **14.33** | **70.28** | 0.43 |
> | PRISM-lite ViT-S | **78.97** | **69.60** | **84.71** | 14.39 | 69.73 | **0.61** |
>
> These results suggest that even **PRISM-lite** remains highly competitive: it achieves a higher $\Delta_m$ than SAK (**0.61 vs. 0.43**) and is stronger on semseg/parsing/saliency. The weaker normals/boundary performance aligns with our known trade-off. We will therefore revise our claim to be more measured: PRISM is **not** universally cheaper, but trades higher **offline** distillation cost for a **single deployable student**.
>
> ### w5. Sensitivity analysis / robustness of routing patterns
> We added broader sensitivity results on ViT-S/PASCAL-Context over Top-k, expert capacity **N**, the shared expert, and the depth of $L_{decorr}$ (**see R4(Bwop), w1**), showing that PRISM is not brittle and that the default gives the best overall trade-off.
>
> ### q3. Are the specialization patterns stable across different random seeds?
> **Yes, the patterns are highly stable.** By aligning teacher–expert affinity matrices across checkpoints trained with distinct seeds via Hungarian matching, we found the same routing structure consistently re-emerges as showed in **[Fig.C](https://anonymous.4open.science/r/anonymous_rebuttal/README.md)**: teacher-preferred experts remain distinct in early/mid layers, while the deepest layer becomes shared. This confirms the specialization is not a one-time artifact.

---

> > ### Author Rebuttal · Reviewer_Pt7x · 2026-04-05
> >
> > Most of questions/concerns are addressed, I am updating the score to reflect this

---

> > > ### Author Response · Authors · 2026-04-06
> > >
> > > We sincerely thank the reviewer for the constructive feedback and for the positive acknowledgment after reading our rebuttal. We are very glad that the additional clarifications and evidence helped resolve the concerns, and we will reflect these clarifications more clearly in the revised manuscript, particularly regarding the practical motivation, relation to prior pre-VFM multi-teacher distillation literature, PRISM's unique capability, and the empirical discussion of the trade-off between cost and performance and routing robustness.

---

### Official Review · Reviewer_96sY · 2026-03-12

**Soundness:** 2
**Presentation:** 3
**Significance:** 3
**Originality:** 2
**Overall Recommendation:** 4
**Confidence:** 4

**Summary:**

This paper proposes PRISM, a dual-stream Mixture-of-Experts framework for multi-teacher VFM distillation. The key insight is that standard dense architectures suffer gradient conflict when distilling contradictory teachers. PRISM decomposes knowledge into orthogonal expert subspaces via context-modulated routing, then recombines experts for downstream tasks. A locality-aware decorrelation loss prevents routing collapse. Experiments on PASCAL-Context and NYUD-v2 show state-of-the-art multi-task dense prediction performance.

**Compliance With Llm Reviewing Policy:**

Affirmed.

**Final Justification:**

Thank you for the detailed rebuttal. The additional analyses improve the empirical support, particularly regarding routing behavior and cross-teacher interaction. While I still find the mechanism not fully established and the evidence largely correlational, the rebuttal sufficiently strengthens the paper. I therefore revise my rating to 4.

**Key Questions For Authors:**

1) How sensitive is the method to the number of experts and routing configuration?

2) Would PRISM still show benefits when distilling fewer teachers?

3)  Can the authors provide token-level statistics showing how often conflicting teacher gradients for the same token are routed to non-overlapping experts?

**Limitations:**

yes

**Strengths And Weaknesses:**

**Strengths**

1) The paper studies an important and timely problem: integrating knowledge from multiple vision foundation models.

2) The proposed dual-stream architecture provides a reasonable design to balance shared representations and specialized experts.

3) The analysis of gradient conflicts and routing dynamics offers useful insights into how expert specialization emerges.

**Weaknesses**

1) The core theoretical claim that MoE routing achieves gradient orthogonalization lacks token-level verification. Figure 2 reports population-level cosine similarity distributions, which cannot confirm that individual routing decisions consistently separate conflicting gradients for the same token. Without this evidence, the orthogonalization framing remains a plausible narrative rather than a demonstrated mechanism. The narrow two-benchmark evaluation further limits confidence in the claimed generality.

2) The main results are reported on only two benchmarks with a ViT-B/16 backbone. Both datasets are relatively small-scale dense prediction benchmarks. It is unclear whether PRISM's advantages hold on larger-scale datasets, stronger backbones, or tasks beyond dense prediction.

3) Since tokens can activate multiple experts simultaneously, the gradients from different teachers may still interact within the same expert. The paper does not empirically show that the effective inter-task gradient interaction approaches zero as claimed in Equation (3).

4) PRISM achieves a lower overall improvement than SAK (10.59 vs. 11.11 Δm). In particular, the gap on Boundary Detection (76.59 vs. 78.60) is substantial. The paper attributes this to the lack of dedicated adapters but does not investigate whether this reflects a fundamental limitation of the proposed architecture.

---

> ### Author Rebuttal · Authors · 2026-03-31
>
> We sincerely thank you for recognizing our work's importance, design, and insights. Regarding Eq. (3), we acknowledge our original phrasing was too strong. Exact gradient orthogonalization is our design objective, not an absolute analytical guarantee for every token under top-k routing. We have revised our claim to be more precise: PRISM **substantially reduces effective cross-teacher interference in practice**. To address your core concern, we present the requested token-level routing statistics below.
>
> ### w1, w3 & q3. Token-level verification of routing orthogonalization
> **(1)** We first ask whether the **same token** is routed to the same sparse experts under different teacher conditions. Here **C=CLIP, D=DINOv2, S=SAM**; e.g., **C-D** denotes the mean token-level Jaccard overlap between the top-3 expert sets selected under CLIP vs. DINOv2 conditions.
>
> | Layer | C-D ↓ | C-S ↓ | D-S ↓ |
> |-|-:|-:|-:|
> | 2  | 0 | 0 | 0 |
> | 5  | .0012 | 0 | 0 |
> | 8  | .0011 | 2e-6 | 0 |
> | 11 | .0034 | .0093 | .0822 |
>
> Overlap is essentially zero in early/mid layers and remains small even at L11 (max **.0822**), showing that the **same token** is routed to almost disjoint sparse experts under different teacher conditions.
>
> **(2)** We next test whether PRISM separates routing more strongly on tokens where two teachers disagree more. For each patch token, we compare the **most conflicting 25%** and **least conflicting 25%**, where conflict is defined by whether the two teachers would push the same student token in similar or conflicting directions.
>
> We first verify that this split is meaningful: across all **12 pair×layer cases**, the least-conflicting group always has higher cross-teacher gradient similarity than the most-conflicting group. We then ask whether this is reflected in routing. In later-layer cases where routing overlap is non-trivial, the most conflicting tokens show lower sparse-expert overlap in **4/5** cases, e.g., DINOv2-CLIP at L11 (**.0920 → .0847**) and CLIP-SAM at L11 (**.1013 → .0980**). Earlier layers are already nearly fully separated, so both groups have near-zero overlap.
>
> Thus, PRISM does not only separate teachers globally; it tends to assign more disjoint sparse routes to tokens with stronger cross-teacher conflict.
>
> **(3)** We also directly measure gradient cosine on the actually co-active sparse experts across checkpoints from the same training run. As shown in **[Fig.A / Fig.B](https://anonymous.4open.science/r/anonymous_rebuttal/README.md)**, the number of pair/layer cases with **no co-active sparse experts** increases from **4/12 (early)** to **7/12 (mid/late/final)**. For a representative case (D-C at L5), the number of co-active experts decreases from **9 → 7 → 2 → 2**, while the sparse expert-gradient cosine drops from **.235 → .189 → -.167 → -.166**.
>
> Therefore, top-k routing does not provide strict zero interaction analytically, but in practice it often yields **zero co-activation**; and when co-activation remains, the residual interaction becomes much smaller and can even turn negative.
>
> ### w2. Limited evaluation / stronger backbones / tasks beyond dense prediction
> Our current evaluation follows prior comparable dense-prediction setups (e.g., SAK, MLoRE, BFCI) for fair comparison. We agree that broader validation would further strengthen the paper; due to rebuttal-time constraints, we can only provide a **preliminary ViT-L** result here (see **R3(siEV), w4**).
>
> ### w4. Boundary gap vs. SAK
> Boundary prediction requires high-frequency local cues; semantic tasks need global abstraction. SAK and MLoRE excel here as isolated adapters preserve local signals. Under limited capacity (ViT-S/B), PRISM prioritizes semantic disentanglement. Removing $\mathcal{L}_{\text{decorr}}$ slightly improves boundary (70.78→70.86) but severely hurts parsing (69.25→61.80), showing a multi-task trade-off. This gap stems from capacity limits, not mechanism: our ViT-L (10% data) achieves better boundary (76.36 vs SAK's 76.27) and Normal (13.49 vs 13.82), supportting that PRISM can captures both global and local cues  given sufficient capacity. We will complete the full ViT-L experiments, and include in the revision.
>
> ### q1. Sensitivity to the number of experts and routing configuration
> We added broader sensitivity results on ViT-S/PASCAL-Context over Top-k, expert capacity **N**, the shared expert, and the depth of $L_{decorr}$ (**see R4(Bwop), w1**), showing that PRISM is not brittle and that the default gives the best overall trade-off.
>
> ### q2. Would PRISM still help with fewer teachers?
> Yes. We additionally tested a **reduced-teacher setting using DINO + SAM**. PRISM still clearly outperforms a dense widened student, so the benefit does not rely on the full teacher set.
>
> | Model | Semseg ↑ | Parsing ↑ | Saliency ↑ | Normal ↓ | Boundary odsF ↑ |
> |---|---:|---:|---:|---:|---:|
> | Wide ViT | 66.13 | 61.21 | 84.16 | 14.73 | 67.08 |
> | **PRISM** | **77.80** | **68.90** | **84.50** | **14.43** | **69.48** |

---

> > ### Author Rebuttal · Reviewer_96sY · 2026-04-03
> >
> > Thank you for the rebuttal. I appreciate that the authors directly addressed my main concern about the orthogonalization claim. The added token-level routing overlap statistics and co-active expert analysis are helpful, and the clarification that exact orthogonalization is a design objective rather than a strict guarantee makes the claim more precise.
> >
> > That said, I still think some concerns remain only partially resolved. First, while the new routing statistics make the mechanism more plausible, they still do not fully characterize effective cross-teacher gradient interaction across teacher pairs, layers, and checkpoints. Second, the broader generality of the method remains unclear, since the main evaluation is still limited to two dense prediction benchmarks, and the larger-backbone evidence is still preliminary rather than directly comparable. Third, the training cost is now clearer, but the practical cost-performance trade-off versus SAK is still not fully established. Finally, the additional regularization comparisons are helpful, but the role of the decorrelation loss could still be explained more clearly, especially why its effect appears particularly strong on Human Parsing.
> >
> > Overall, the rebuttal improves the paper and addresses part of my concerns, but some important issues remain unresolved, so my overall evaluation remains unchanged.

---

> > > ### Author Response · Authors · 2026-04-06
> > >
> > > We are glad that our response on orthogonalization claim and token-level routing overlap statistics have addressed your concerns. And thank you for your follow up questions regarding effective cross-teacher gradient interaction, broader generality and practical cost-performance trade-off versus SAK.
> > >
> > > Here we provide additional clarification and targeted evidence on the remaining points.
> > >
> > > ### q1: effective cross-teacher gradient interaction
> > >
> > > We provide a complete 3 pair × 4 layer × 4 checkpoint characterization of effective cross-teacher interaction.
> > >
> > > For each teacher pair, MoE layer, and checkpoint from the same training run, we measure the interaction on the actually co-active sparse experts:
> > >
> > > $I_{l,t}^{(a,b)} = E\\!\left[1(E_l^a \\cap E_l^b \\neq \\emptyset) \\cdot \\frac{1}{|E_l^a \\cap E_l^b|} \\sum_{e \\in E_l^a \\cap E_l^b} \\cos(g_{l,e}^a, g_{l,e}^b)\\right]$
> > >
> > > which is exactly zero when no sparse expert is co-activated, and otherwise directly quantifies the residual gradient interaction on the updated expert parameters.
> > >
> > > Specifically, we report: **(a)** the number of co-active sparse experts, **(b)** token-level routing overlap, **(c)** the gradient cosine on the actually co-active sparse experts, and **(d)** the gradient cosine on the shared expert, in **[Fig. F](https://anonymous.4open.science/r/anonymous_rebuttal/anonymous_rebuttal_2.md)**.
> > >
> > > The results show that **PRISM reduces sparse-path interference through two complementary regimes**. In many pair-layer cases, routing overlap and co-activation decrease sharply during training; for example, for DINOv2-CLIP at L5, token overlap drops from 0.520 to 0.018, co-active experts from 8.9 to 2.0, and sparse-expert gradient cosine from 0.107 to 0.002. In deeper cases, **even when routing overlap persists, the residual sparse-expert gradient cosine remains near zero**; most notably, DINOv2-SAM at L11 keeps high overlap, yet its sparse-expert gradient cosine stays around 0 throughout training. Importantly, the shared expert retains small but nonzero compatible gradients, showing that PRISM does not suppress all interaction, but instead learns a structured decomposition: the sparse path mitigates teacher-specific interference, while the shared component preserves the remaining compatible signal.
> > >
> > > ### q2: the broader generality of the method
> > > Our current evaluation follows the standard protocol used by prior comparable multi-teacher dense prediction methods. To further address the reviewer’s concern about broader generality and stronger-backbone evidence, we include a **full ViT-L** experiment with **Stage 1: ImageNet-1K pretraining** and **Stage 2: PASCAL-Context fine-tuning**, following the same setup as **SAK ViT-L**:
> > >
> > > | Configuration | Semseg ↑ | Parsing ↑ | Saliency ↑ | Normals ↓ | Boundary odsF ↑ | $\Delta_m$ ↑ |
> > > |-|-:|-:|-:|-:|-:|-:|
> > > | Single-task baseline ViT-L | 81.61 | 72.77 | 83.80 | 13.87 | 75.24 | 0.00 |
> > > | SAK ViT-L  | 84.01 | 76.99 | 84.65 | 13.82 | **76.27** | 2.30 |
> > > | **PRISM ViT-L** | **84.34** | **77.83** | **84.67** | **13.43** | 76.23 | **3.16** |
> > >
> > > This result further supports the generality of PRISM: when scaled to ViT-L, it attains a stronger overall $\Delta_m$ than SAK.
> > >
> > > ### q3: the practical cost-performance trade-off versus SAK
> > > We agree that the practical trade-off should be stated carefully. Due to the rebuttal length limit, the full discussion is provided in **R2(Pt7x), w4**; here we summarize the key evidence.
> > >
> > > We report a matched **ViT-S** comparison using **PRISM-lite**, which keeps the **same PRISM framework, routing design, losses, and training protocol** as the full PRISM model, and changes only one capacity knob: the **expert MLP ratio is reduced from \(4.0\) to \(1.0\)**. This isolates model capacity rather than changing the method itself.
> > >
> > > | Model | Total Params (M) ↓ | Active Params (M) ↓ | GFLOPs ↓ | Latency (ms) ↓ |
> > > |-|-:|-:|-:|-:|
> > > | SAK-ViT-S | 26.40 | 26.40 | 53.20 | 20.25 |
> > > | PRISM-lite ViT-S | 48.05 | 38.06 | 57.51 | 47.51 |
> > >
> > > | Model | Semseg ↑ | Parsing ↑ | Saliency ↑ | Normal ↓ | Boundary odsF ↑ | $\Delta_m$ ↑ |
> > > |-|-:|-:|-:|-:|-:|-:|
> > > | SAK ViT-S | 78.66 | 68.46 | 84.66 | **14.33** | **70.28** | 0.43 |
> > > | PRISM-lite ViT-S | **78.97** | **69.60** | **84.71** | 14.39 | 69.73 | **0.61** |
> > >
> > > These results support a more precise conclusion: PRISM is indeed not uniformly cheaper than SAK, but still at the same cost level; it trades higher **offline** distillation cost for a **single deployable student** that consolidates heterogeneous teacher knowledge, while remaining competitive even in a lighter form. Notably, PRISM-lite still achieves a higher overall $\Delta_m$ than SAK.
> > >
> > > ### q4: Why the effect of $L_{decorr}$ appears particularly strong on Human Parsing
> > >
> > > We address this point separately in the newly added reply to **R3(siEV)** due to the rebuttal length limit. There we provide a focused explanation and additional evidence on why the effect of $L_{decorr}$ is particularly visible on Human Parsing.

---

### Decision · Program_Chairs · 2026-04-30

**Decision:**

Accept (regular)

**Comment:**

At the end of the author-reviewer discussion period, two of the reviewers confirmed that their original concerns were fully addressed and the other two noted that their issues were only partially addressed. Upon further discussion, and with additional responses from the authors, one of these two reviewers (siEV) found the responses sufficient to recommend acceptance. Although the other reviewer (96sY) noted some drawbacks, they do not seem to pose a major hindrance for acceptance. Thus, on balance, the AC recommends the paper for acceptance.